# Terrestrial-type nitrogen-fixing symbiosis between seagrass and a marine bacterium

Wiebke Mohr[1]✉, Nadine Lehnen[1], Soeren Ahmerkamp[1], Hannah K. Marchant[1], Jon S. Graf[1], Bernhard Tschitschko[1], Pelin Yilmaz[1,4], Sten Littmann[1], Harald Gruber-Vodicka[1], Nikolaus Leisch[1], Miriam Weber[2], Christian Lott[2], Carsten J. Schubert[3], Jana Milucka[1] & Marcel M. M. Kuypers[1]

Symbiotic $N_2$-fixing microorganisms have a crucial role in the assimilation of nitrogen by eukaryotes in nitrogen-limited environments[1–3]. Particularly among land plants, $N_2$-fixing symbionts occur in a variety of distantly related plant lineages and often involve an intimate association between host and symbiont[2,4]. Descriptions of such intimate symbioses are lacking for seagrasses, which evolved around 100 million years ago from terrestrial flowering plants that migrated back to the sea[5]. Here we describe an $N_2$-fixing symbiont, 'Candidatus Celerinatantimonas neptuna', that lives inside seagrass root tissue, where it provides ammonia and amino acids to its host in exchange for sugars. As such, this symbiosis is reminiscent of terrestrial $N_2$-fixing plant symbioses. The symbiosis between Ca. C. neptuna and its host Posidonia oceanica enables highly productive seagrass meadows to thrive in the nitrogen-limited Mediterranean Sea. Relatives of Ca. C. neptuna occur worldwide in coastal ecosystems, in which they may form similar symbioses with other seagrasses and saltmarsh plants. Just like $N_2$-fixing microorganisms might have aided the colonization of nitrogen-poor soils by early land plants[6], the ancestors of Ca. C. neptuna and its relatives probably enabled flowering plants to invade nitrogen-poor marine habitats, where they formed extremely efficient blue carbon ecosystems[7].

Seagrasses form vast meadows in coastal environments around the globe, providing a breeding ground and food for fish and protection from coastal erosion[8–10]. Furthermore, seagrass meadows have a major role in the drawdown of carbon dioxide ($CO_2$) due to their large biomass production, which matches that of the most prolific terrestrial ecosystems[11]. The nitrogen (N) that is required for this biomass production is generally believed to be taken up by the seagrasses through leaves and roots from the surrounding environment[12]. As many seagrasses are found in oligotrophic, N-depleted environments, the seagrass productivity is thought to be at least partially supported by N originating from microbial $N_2$ fixation[13,14]. The $N_2$ fixation is generally assumed to take place in the surrounding sediment by microorganisms that are associated with either the rhizosphere/rhizoplane[14,15] or with animals residing in the seagrass meadows[16]. By contrast, terrestrial plants that thrive in N-poor habitats often enter more intimate, mutually beneficial interactions with $N_2$-fixing bacteria[2,17,18]. The bacteria usually reside within the plant tissue, and the interaction between these symbionts and their plant hosts is genetically complex[19], relying on a sophisticated communication and metabolite exchange[20]. Here, we report the discovery of a marine $N_2$-fixing bacterium that lives inside the root tissue of the seagrass P. oceanica, exhibiting features that are reminiscent of terrestrial plant $N_2$-fixing symbionts.

## Growth and $N_2$ fixation in seagrass meadows

P. oceanica from the oligotrophic Mediterranean Sea is one of the most productive seagrasses[21]. At our study site, P. oceanica forms dense meadows with around 600 shoots per $m^2$ (Fig. 1a). In situ measurements taken during summer 2019 revealed that these meadows had high rates of photosynthesis, resulting in a net primary production of around 50 mmol $m^{-2}$ $d^{-1}$ $CO_2$ fixed (Fig. 1b and Extended Data Fig. 1). The primary production was comparable to that reported for other P. oceanica meadows in the Mediterranean Sea[11]. By contrast, the non-vegetated sandy sediments surrounding the meadows were a net source of $CO_2$, releasing ~6 mmol $m^{-2}$ $d^{-1}$ $CO_2$, despite the presence of benthic algal biofilms (Fig. 1b and Extended Data Fig. 1). The high primary production associated with the P. oceanica meadows occurred in the absence of detectable nutrient-N in the water column (Extended Data Table 1a).

Our $^{15}N_2$-labelling experiments with roots and rhizomes of whole plants (Fig. 1c) revealed seasonal $N_2$ fixation activity that was mainly associated with the roots of P. oceanica (Fig. 1e). The rates of root-associated $N_2$ fixation were highest during summer, when inorganic-N concentrations in the water column were below the detection limit (Figs. 1e and 2a and Extended Data Table 1a). In spring, when inorganic N was detectable, $N_2$ fixation rates were low to non-detectable (Fig. 1e and Extended Data Table 1a). Although the $^{15}N_2$-labelling experiments were

[1]Max Planck Institute for Marine Microbiology, Bremen, Germany. [2]HYDRA Marine Sciences GmbH, Bühl, Germany. [3]Swiss Federal Institute of Aquatic Science and Technology (Eawag), Department of Surface Waters-Research and Management, Kastanienbaum, Switzerland. [4]Present address: Data Science Research Group, Institute for Artificial Intelligence in Medicine, University Hospital Essen, Essen, Germany. ✉e-mail: wmohr@mpi-bremen.de

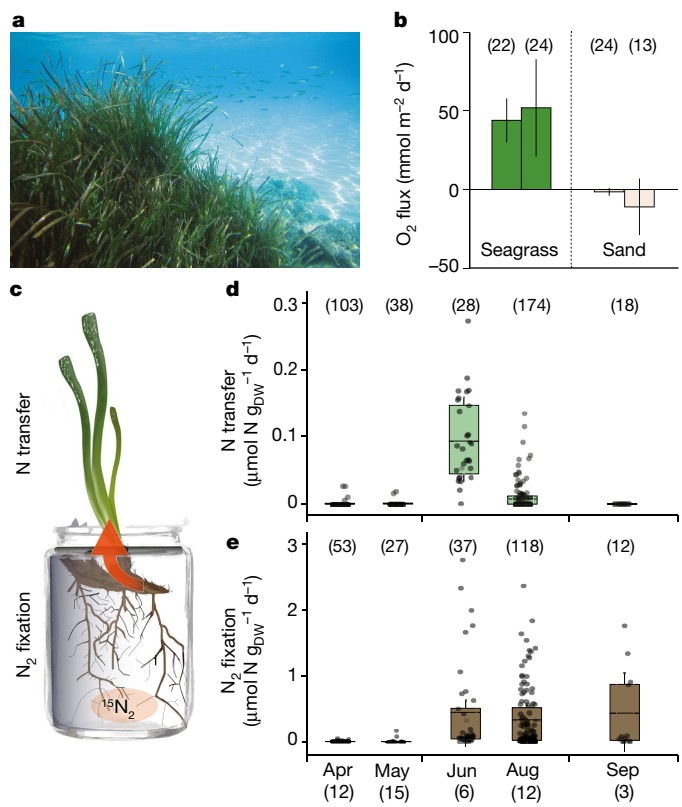

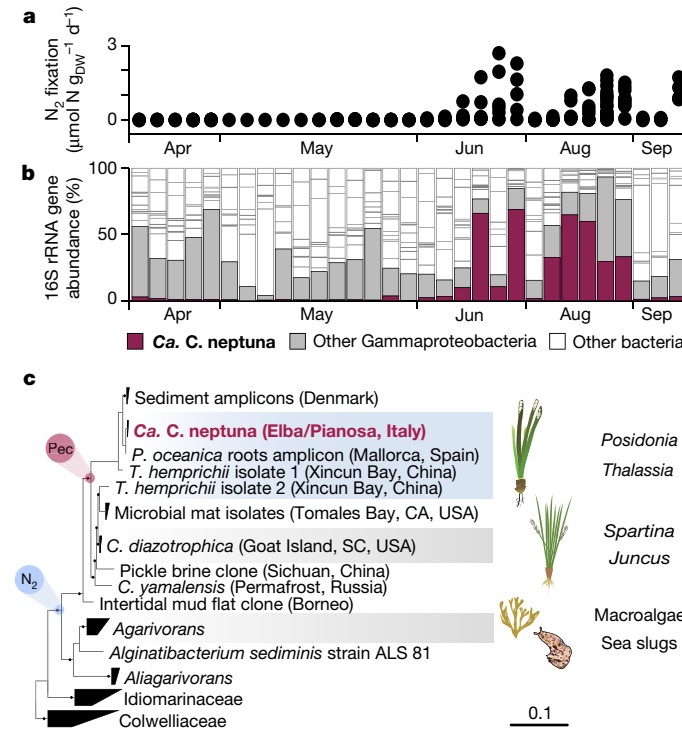

**Fig. 1 | Productivity, root-associated N₂ fixation and N transfer to leaves.**
**a**, *P. oceanica* meadow in the sandy sediments of Fetovaia Bay, Elba (Italy). **b**, Areal net O₂ fluxes in the seagrass meadow (green) and neighbouring non-vegetated sandy sediments (light brown). The four individual bars represent the daily mean values of averaged hourly fluxes, and the error bars indicate the variability of night-time and daytime fluxes (propagated s.d.). The four measurements, two seagrass (June 2019) and two sand (June 2019 and September 2018), were performed on four different days (see Extended Data Fig. 1 for examples of a daily cycle). **c**, Illustration of the incubation set-up enabling the detection of the transfer of freshly fixed (¹⁵N-enriched) N from the roots to the leaves (orange arrow). **d**, **e**, Root-associated N₂ fixation (**e**) and N transfer to the leaves (**d**) of *P. oceanica* from April to September. The bars, boxes and error bars represent the mean values, 25th and 75th percentiles, and s.e.m. The number of measured plant pieces (*n*) is indicated in parentheses above the data points with the number of incubated plants indicated below each month. DW, dry weight. Note the tenfold difference in scale between **d** and **e**.

restricted to roots and rhizomes, leaf biomass was also enriched in ¹⁵N in summer, indicating that newly fixed N was transferred from the roots to the leaves (Fig. 1d). This transfer was rapid, with up to around 20% of the freshly fixed N being assimilated into leaf biomass already within 24 h (Fig. 1d). Such rapid transfer to the leaves was previously reported for *Zostera marina*[22]. Taking into account all fixed N recovered in the different plant organs, root-associated N₂ fixation could fully support the measured in situ net plant biomass production in summer (Fig. 1b and Supplementary Information), which is the main growth season of *P. oceanica*[23]. Furthermore, root-associated N₂ fixation is probably a source of N for the wider seagrass ecosystem, as indicated by elevated inorganic N concentrations in seagrass sediments relative to the surrounding, non-vegetated sediments (Extended Data Table 1b).

## An N₂-fixing root endophyte

Metagenomic sequencing revealed substantial differences between the microbiome of *P. oceanica* roots and the surrounding sediments (Extended Data Fig. 2), indicating that, like other seagrasses[24], *P. oceanica* also has a specialized root microbiome. Moreover, on the

**Fig. 2 | Relative abundance and phylogeny of *Ca*. C. neptuna. a**, The root-associated N₂ fixation rates of individual plants. Each symbol represents an individually measured root piece. **b**, The relative abundance of *Ca*. C. neptuna 16S rRNA gene sequence reads (magenta) in roots of individually analysed plants. One column represents one plant and each column corresponds to the measured N₂ fixation rates in **a**. **c**, Phylogeny of *Ca*. C. neptuna (bold, magenta) within the Celerinatantimonadaceae based on 16S rRNA gene sequences (consensus tree; the scale bar is substitutions per site; the black dots indicate strong bootstrap support). Macrophytes and metazoa from which sequences were recovered are indicated on the right. Blue and magenta circles indicate the acquisition of abilities to fix N₂ and to degrade pectin (Pec), respectively. *C. yamalensis*, *Celerinatantimonas yamalensis*; *T. hemprichii*, *Thalassia hemprichii*. Accession numbers and references are provided in Supplementary Data 1 and 2.

basis of 16S rRNA amplicon data, the root microbial communities of N₂-fixing plants (Methods) differed substantially from plants with non-detectable N₂ fixation rates (Fig. 2b and Extended Data Fig. 3). The difference was largely driven by a single gammaproteobacterium belonging to the genus *Celerinatantimonas*, which was abundant in the roots of N₂-fixing plants and relatively rare in non-N₂-fixing plants (Fig. 2b and Extended Data Fig. 3). The closest cultured relative was *Celerinatantimonas diazotrophica* (~95% 16S rRNA gene similarity), an N₂-fixing bacterium isolated from saltmarsh grasses[25] (Fig. 2c). On the basis of thresholds for genus discernment[26], the bacterium recovered from *P. oceanica* roots represents a new species within the genus *Celerinatantimonas* (Supplementary Information), which we named *Candidatus* Celerinatantimonas neptuna (*Ca*. C. neptuna).

Specific 16S rRNA-targeted probes were designed to visualize *Ca*. C. neptuna cells in root sections using fluorescence in situ hybridization (FISH). Few *Ca*. C. neptuna cells were found inside the roots of non-N₂-fixing plants from spring (Extended Data Fig. 4). By contrast, endophytic *Ca*. C. neptuna cells were abundant (~80% of 4′,6-diamidino-2-phenylindole (DAPI) counts) throughout the root cortex and stele of N₂-fixing plants in summer (Fig. 3a–c and Extended Data Figs. 4 and 5). *Ca*. C. neptuna cells resided in the intercellular spaces as well as inside plant root cells (Fig. 3b, c and Extended Data Fig. 5). Single-cell measurements using nanoscale secondary ion mass spectrometry (nanoSIMS) provided direct evidence that *Ca*. C. neptuna

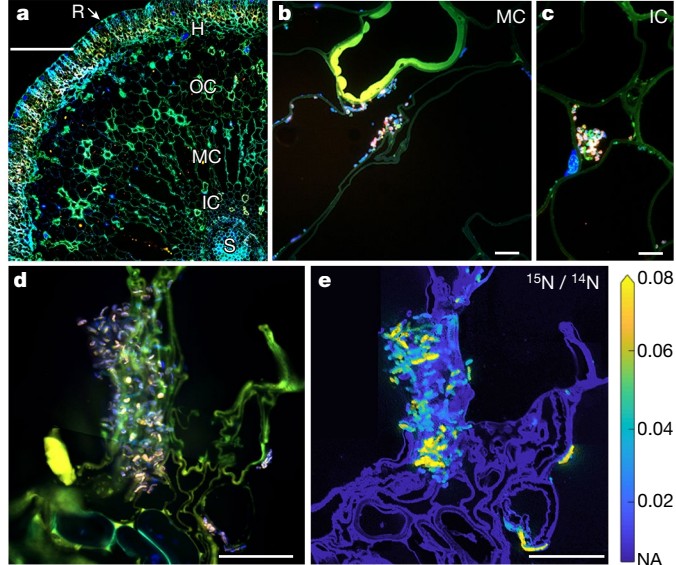

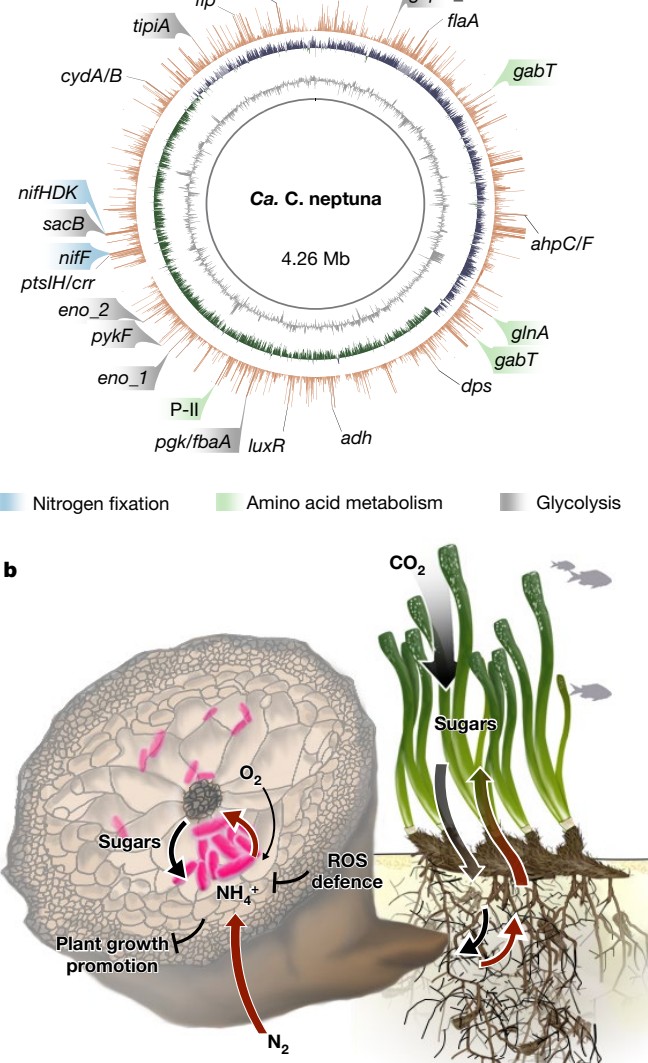

**Fig. 3 | Distribution and N₂ fixation activity of *Ca*. C. neptuna. a**, Epifluorescence image (stitched) of a root cross-section (overlay image of DAPI (blue) and autofluorescence (green/orange)). **b**, **c**, Images of *Ca*. C. neptuna cells and clusters (pinkish colour due to the overlap of the DAPI and FISH probe (orange) signals) in the middle cortex (**b**) and inner cortex (**c**). **d**, **e**, Correlative imaging (stitched images) of *Ca*. C. neptuna cells in the inner and middle cortex (**d**) and the corresponding nanoSIMS image showing $^{15}N$ enrichment (**e**). H, hypodermis; IC, inner cortex; MC, middle cortex; NA, natural abundance; OC, outer cortex; R, rhizoplane; S, stele. Scale bars, 150 μm (**a**) and 10 μm (**b**–**e**).

fixed $^{15}N_2$ in the roots of *P. oceanica* in summer (Fig. 3d, e and Extended Data Fig. 4). The *P. oceanica* root tissue was also substantially enriched in $^{15}N$, indicating that a substantial amount of freshly fixed N (up to ~98%) was transferred to the seagrass (Extended Data Fig. 4 and Supplementary Information).

## Metabolic capacity of the seagrass endophyte

To gain insights into the metabolic interaction between *Ca*. C. neptuna and *P. oceanica*, we obtained the genome and transcriptome of *Ca*. C. neptuna from N₂-fixing plants. The ~4.3 Mb metagenome-assembled genome of *Ca*. C. neptuna comprised a single circular chromosome encoding all of the enzymes necessary for N₂ fixation (Extended Data Fig. 6). Genes coding for nitrogenase, the enzyme that catalyses the reduction of N₂ to ammonium, as well as proteins that transfer electrons to the nitrogenase (that is, *nifHDK/F*) were highly transcribed under N₂-fixing conditions (Fig. 4a and Extended Data Figs. 6 and 7). Some of the ammonium produced by *Ca*. C. neptuna was probably directly transferred to the seagrass (Fig. 4b). Furthermore, in analogy to terrestrial N₂-fixing plant symbioses[27], fixed N was also transferred in the form of amino acids. Glutamate, phenylalanine and leucine were probably transferred from *Ca*. C. neptuna to the seagrass roots, as indicated by the incorporation of $^{15}N$ into these protein-bound amino acids (Extended Data Fig. 8 and Supplementary Information). In return, the seagrass may provide the amino acid GABA (4-aminobutyrate) or precursors (such as arginine or ornithine), analogous to some rhizobia–legume symbioses[27–29]. Correspondingly, the *gabT* gene, which encodes an aminotransferase that catalyses the amino-group transfer from GABA to 2-oxoglutarate to yield glutamate, was among the most highly transcribed genes (Fig. 4a and Extended Data Fig. 6).

In addition to GABA, the seagrass probably provides sugars, based on the high transcription of genes encoding extracellular sucrose degradation (*sacB*), sugar-transport proteins (*ptsIH/crr*) and enzymes of the glycolysis pathway (*gapA1*, *pgk*, *eno1*, *eno2*, *pykF*, *fbaA*, *tpiA*) (Fig. 4a

**Fig. 4 | Highly transcribed genes in *Ca*. C. neptuna and the proposed metabolic interaction between *Ca*. C. neptuna and *P. oceanica*. a**, The circular *Ca*. C. neptuna genome with GC content (grey), GC skew (purple/green) and the average transcription of protein-coding genes plotted as transcripts per million (TPM) (orange; TPM values above 1,000 were cut off). Note that most of the highlighted genes related to key metabolic functions have average TPM values of >1,000. A list of gene names and corresponding enzymes is provided in the Supplementary Information. **b**, Schematic of the symbiotic interaction between *Ca*. C. neptuna (magenta) and *P. oceanica* indicating the transfer of fixed N from N₂ fixation (dark red arrows) and plant-derived sugars (black arrows); the potential for further plant growth promotion and defence mechanisms is also indicated. ROS, reactive oxygen species.

and Extended Data Fig. 7). Although genes involved in the uptake of dicarboxylic acids (*dctPQM*) were only moderately transcribed, *Ca*. C. neptuna might also receive $C_4$-dicarboxylates from its host, analogous to Rhizobia[20]. Sugar oxidation in *Ca*. C. neptuna might proceed under microoxic and partly anoxic conditions as indicated by the low transcription of genes encoding the low-O₂-affinity *bo*-type terminal oxidase (*cyoABCDE*), high transcription of genes encoding the high-O₂-affinity *bdI*-type terminal oxidase (*cydAB*) and proteins involved in fermentation (*adh*, *pflB*) (Fig. 4a and Extended Data Fig. 7). Microoxic/anoxic conditions might easily develop in *P. oceanica* roots, which reside in anoxic sediments at our study site (Extended Data Fig. 1c) and other sites throughout the Mediterranean[30]. Under such conditions, root

endophytes depend on their host for oxygen supply, which may enable *P. oceanica* to regulate the proliferation of *Ca*. C. neptuna similarly to the manner in which legume hosts control Rhizobial growth[31]. Microoxic conditions would be favourable for the activity of the oxygen-sensitive nitrogenase of *Ca*. C. neptuna.

In many aspects, the genome of *Ca*. C. neptuna exhibits hallmarks of a facultative endophytic symbiont. Just like many terrestrial plant endophytes, *Ca*. C. neptuna might switch between free-living and host-associated stages[32]. Genes related to motility and attachment (*flaA* and *flp*) were highly transcribed (Fig. 4a and Extended Data Fig. 6), indicating active invasion and colonization of seagrass root tissue. High transcription of the quorum-sensing master regulator *luxR* indicates cell-to-cell communication and orchestration of *Ca*. C. neptuna population activity, which is also important for the establishment of rhizobia-legume symbioses[33]. Furthermore, high transcription of genes related to peroxide detoxification (*dps*, *ahpC/F*) by *Ca*. C. neptuna (Fig. 4a and Extended Data Fig. 6) indicates that reactive oxygen species are produced by seagrasses as a defence mechanism that is comparable to responses by legume hosts[34]. The genome also contains genes that are commonly found in endophytes[35], which are used in host–symbiont recognition, chemotaxis, degradation of plant cell-wall components, plant growth promotion and effector secretion (Supplementary Information). Although many of these traits are not unique to plant-beneficial and/or endophytic microorganisms[36], many are deemed to be vital for establishing a beneficial association[37,38]. On the basis of our combined results, *Ca*. C. neptuna is a plant-beneficial N$_2$-fixing endophyte (Fig. 4b and Extended Data Fig. 9) that is strikingly similar to those in terrestrial plants[2,17,18].

## Acquisition of a marine N$_2$-fixing symbiont

Seagrasses evolved around 100 million years ago[5] from terrestrial flowering plants that migrated back to the sea, where they had to adjust their physiology to a fully submerged lifestyle in salt water[39]. During the transition to a marine plant, the root-associated microbiome of terrestrial origin was probably replaced by a marine one. Accordingly, many of the root-associated microorganisms of *P. oceanica* are typical marine benthic bacteria, such as sulfate reducers and sulfide oxidizers (Supplementary Information).

We can only speculate when and where *P. oceanica* acquired its marine N$_2$-fixing symbiont *Ca*. C. neptuna. The phylogeny of *Ca*. C. neptuna suggests that its ancestor was obtained in a coastal marine environment. Interestingly, the closest relative of *Ca*. C. neptuna—*C. diazotrophica*—was isolated from the roots of the saltmarsh grasses *Juncus* and *Spartina* (Fig. 2c). Moreover, 16S rRNA gene sequences belonging to another member of the genus *Celerinatantimonas* were recovered from the seagrass *Thalassia*. These two plant-associated members of the genus *Celerinatantimonas* may form symbioses with their hosts similar to the symbiosis that *Ca*. C. neptuna forms with its host, but a confirmation of their lifestyle is so far lacking. Members of the more deeply branching genera *Agarivorans*, *Aliagarivorans* and *Alginatibacterium*, which also belong to the family Celerinatantimonadaceae (Supplementary Information), have been found to be associated with macroalgae (Fig. 2c). Interestingly, the potential to fix N$_2$ is prevalent among the members of the family Celerinatantimonadaceae, whereas this potential is missing from known representatives of the neighbouring families Idiomarinaceae and Colwelliaceae (Supplementary Data 3), which typically do not associate with macrophytes. Thus, the last common ancestor of the family Celerinatantimonadaceae was probably associated with macroalgae, and the ability to fix N$_2$ may have been acquired as an adaptation to living on N-poor carbohydrates. This N$_2$-fixing ancestor diversified and formed a new lineage—the genus *Celerinatantimonas*—that was adapted to living in association with marine flowering plants such as seagrasses and saltmarsh grasses.

It appears that the colonization of seagrass may have required comparably few genomic adaptations as many traits that have been deemed to be vital for the establishment of a beneficial association with flowering plants[35,38] were also found in the genomes of related organisms (Supplementary Data 3). However, a crucial step enabling members of the genus *Celerinatantimonas* to invade marine flowering plants was probably the acquisition of a pathway to extracellularly degrade pectin, a polysaccharide that is typically found in terrestrial plant cell walls but is rare in marine algae[40]. The comparison of 34 genomes of related genera and families revealed that only members of the genus *Celerinatantimonas* have the ability to degrade pectin extracellularly (Supplementary Data 3).

Our results show that seagrasses have independently evolved a mechanism to cope with N limitation that is similar to a variety of terrestrial plants[2,17,18]. Just like N$_2$-fixing microorganisms presumably facilitated early land plants to successfully colonize N-poor soil[6], the ancestors of *Ca*. C. neptuna and its relatives probably enabled marine flowering plants to invade and thrive in N-poor marine habitats, where their descendants form the basis of extremely efficient blue carbon ecosystems[7].

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

## Methods

### Etymology

'*Candidatus* Celerinatantimonas neptuna' (nep.tu'na L. fem. n.), pertaining to *Neptunus* (L. masc. n. Neptune), the Roman god of the seas and the Neptune grass, *Posidonia oceanica*.

### Sampling

A *P. oceanica* meadow at 8 m water depth and nearby sandy sediments in Fetovaia Bay, Elba, Italy[13] were sampled between June 2014 and September 2019; individual sampling months and years are indicated in the sections below and/or in the figures and tables. In May 2017, a *P. oceanica* meadow at the island of Pianosa, Italy was also sampled. All of the samples were obtained via SCUBA diving.

Complete plants of *P. oceanica* were carefully separated from the meadow by hand and stored in seawater-filled containers until arrival at the shore-based laboratory. Sediment for use in the laboratory-based aquaria was scooped into containers from nearby sandy patches. Seawater was pumped through a hose (placed at about 0.5 m above the *P. oceanica* meadow) into several 50 l barrels onboard the boat and was later used in the laboratory for the aquarium and the incubation experiments.

The sediment within the seagrass meadow was sampled with stainless steel core tubes (length, 50 cm), which were drilled into the sediment by divers, and the cores were briefly stored at 22 °C (ambient temperature, September 2019) in a seawater-filled barrel until further processing at the shore-based laboratory.

Porewater nutrient samples were obtained using stainless steel lances[41] at intervals of around 10 cm. Water column nutrient samples were obtained from above the seagrass meadow at the start or end of sampling. Nutrient samples were collected in 15 ml or 50 ml centrifuge tubes and were stored in a cooler box until further processing.

### Nutrient measurements

Water column nutrients were measured during several sampling campaigns as indicated in Extended Data Table 1a. Ammonium ($NH_4^+$) concentrations were measured fluorometrically[42] in the nearby shore-based laboratory, and the remaining water was frozen (−20 °C) for later analyses of nitrate ($NO_3^-$), nitrite ($NO_2^-$), phosphate ($PO_4^{3-}$) and silicate ($SiO_4^{4-}$) using an autoanalyser (QuAAtro, Seal Analytical). Porewater samples were obtained in June 2019 and were processed the same as the water column nutrient samples with the exception that ammonium was not measured on site but at the home laboratory at the same time as the other nutrients. Dissolved inorganic nitrogen (ammonium plus $NO_x^-$) concentrations in the porewater were averaged for the upper 20 cm (Extended Data Table 1b).

### Net primary production measurements using the EC method

Net carbon dioxide ($CO_2$) fluxes were calculated on the basis of oxygen ($O_2$) fluxes determined using the aquatic eddy covariance (EC) method. In this non-invasive approach, turbulence-induced transport is resolved using high-frequency current meters combined with fast $O_2$ microsensors. Under the assumption of stationarity, the instantaneous turbulent flux contributions are calculated by correlating vertical current fluctuations to oxygen fluctuations. Our EC system was equipped with an acoustic Doppler velocimeter (ADV, Nortek) and ultra-fast responding optode microsensors with a tip diameter of 430 μm ($t_{90} < 0.3$ s, Pyroscience). The microsensor tip was aligned vertically to the centre of the ADV measuring volume and shifted 2.5 cm horizontally to avoid current disturbances. Inside the seagrass meadow, the EC system was installed upside-down, such that the measuring volume was approximately 40 cm above the seagrass canopy height, while an upright installation was chosen for sandy sediments with the measuring volume about 20 cm above the sea floor. Additional sensors were used to monitor long-term $O_2$ changes (Aanderaa, 4831),

temperature variations (PT100, custom made) and photosynthetic active radiation sensor (Biospherical Instruments, QCP-2000). All of the instruments were attached to an aluminium frame, which enabled SCUBA-diver-operated deployments. Current data were recorded at 16 Hz and $O_2$ data at 1–4 Hz, limited by the response time of the sensor. All of the other instruments recorded at 0.1 Hz.

The data were processed according to standard procedures for aquatic eddy correlation measurements[43,44] in MATLAB 2018b (Mathworks). First, current data were downsampled to the frequency of $O_2$ measurements, despiked and corrected for the tilt of the ADV. Subsequently, current and $O_2$ data were decomposed into the steady and fluctuating component using a low-pass filter. The fluctuating $O_2$ time series was shifted until a maximum correlation with the vertical velocity fluctuations was achieved; this was typically in the range of 2–3 s resulting from low horizontal velocities. Instantaneous fluxes are usually highly variable; thus, two averaging procedures were applied: (1) instantaneous fluxes were temporally integrated to determine cumulative fluxes, (2) if the cumulative fluxes showed sudden jumps indicating erroneous measurements, the time series was truncated at that point. Instantaneous fluxes were then averaged for a 60 min burst and were subsequently averaged over the entire measurement time. Negative fluxes during the night represent respiration, whereas positive fluxes during the day represent the sum of respiration and gross primary production. In June 2019, the EC system was deployed twice for 22–24 h in the centre of the sampled seagrass meadow. Seagrass metabolism rates were referenced by a sandy sediment station (24 h) at a similar water depth at a distance of approximately 50 m to the meadow sampling site. One additional measurement in sandy sediments was performed for 13 h in September 2018. Net $O_2$ fluxes were converted to net $CO_2$ fluxes using a ratio of 1 mol $O_2$:1 mol $CO_2$. Custom codes for data processing are available (see the 'Code availability' section).

Oxygen concentrations within seagrass sediments were measured using microsensors[45] mounted to the same frame as the EC system.

### Sediment-free incubation of complete plants with the $^{15}N_2$ tracer

During six campaigns (June 2014, May 2015, April 2016, August 2016, May 2017 and September 2018), complete plants of *P. oceanica* were collected by SCUBA diving and were transported to the shore-based laboratory in seawater-filled containers to prevent desiccation. On arrival at the laboratory, the plants were embedded into an aquarium containing sediment that was collected from close to the meadow such that the rhizomes and roots were sediment-covered. The aquarium was equipped with two lamps (OSRAM L58W/77, FLUORA 2250 lm; OSRAM L30W/77, FLUORA 1000 lm) and the site seawater in the aquarium was cooled to ambient temperatures. The plants were kept in the aquarium for up to one day before starting tracer incubations for $N_2$ fixation. To prepare $^{15}N_2$-enriched water for the incubations, site seawater was filtered through a filter (pore size, 0.2 μm) and placed into 0.5 l wide-neck Duran bottles. The filtered seawater was deoxygenated by bubbling with a mixture of nitrogen ($N_2$) and argon (Ar) gas. Concentrations of oxygen ($O_2$) were checked using needle optodes (PreSens, Precision Sensing), and bubbling continued until $O_2$ concentrations were close to zero but at least <10 μmol l$^{-1}$. The nearly anoxic seawater was carefully filled into 60 ml or 120 ml serum bottles, which were crimp-sealed. Then, 10 ml or 20 ml of $^{15}N_2$ gas was injected into the serum bottles in exchange for 5 ml or 10 ml of seawater to produce a slight gas overpressure. The serum bottles were vortexed for around 1 min and left overnight to equilibrate. The $^{15}N_2$ gas (≥99 atomic percentage (at%) $^{15}N$; lot numbers 19197 and 16727, Cambridge Isotopes; purchased from Eurisotop) was tested for $^{15}N$-ammonium contamination[46] using the hypobromite method[47] before use, and no contamination was detected.

To incubate only the roots and rhizomes of complete plants (that is, leaves, rhizome and roots connected) with the $^{15}N_2$ tracer, the water surrounding the roots and rhizomes was separated from that surrounding the leaves (Fig. 1c). We therefore fitted latex gloves (rinsed

three times with filtered seawater) with a sampling port and a little hole through which the leaves of each individual plant were carefully threaded. The latex glove was secured with a clamp on the leaves as close to the meristem as possible. The roots and rhizomes were placed into 0.5 l wide-neck Duran bottles that were filled with deoxygenated water (prepared as described above), and the leaves remained outside the bottle. The glove was used to seal the Duran bottle (together with cable ties and rubber bands). The sampling port enabled the addition of the $^{15}N_2$-enriched water. During all campaigns, except for June 2014, 110 ml of $^{15}N_2$-enriched water was added to each Duran bottle containing roots and rhizomes, without leaving a headspace. Before closing the sampling port, a subsample of the mixed incubation water was taken to measure the enrichment of $^{15}N$ in the $N_2$ pool at the start of the incubation. In June 2014, about 40% of the incubation water was replaced with $^{15}N_2$-enriched water and the enrichment of $^{15}N$ in the $N_2$ pool was measured in the batch of enriched water. Final enrichment was calculated on the basis of the addition of enriched water to the final incubation[13]. The Duran bottle was covered with a black plastic bag to prevent light from reaching the roots and rhizomes. The incubation bottles with the plants were carefully placed into the aquarium. For every experiment, three replicate plants were prepared with the $^{15}N_2$ tracer while one plant was prepared the same way but without the added $^{15}N_2$ tracer (that is, an incubated control for background natural abundance values) except for June 2014 and May 2015, for which natural abundance values were obtained from non-incubated plants at the start of the experiments. Plants were incubated for 4–96 h with most incubations lasting 24 h or 48 h. Incubations were performed as light–dark cycles if incubated for ≥24 h. In June 2014, one additional set of plants was also incubated with 24 h of light. At the end of the incubation, a second subsample of the incubation water was taken through the sampling port to measure the final enrichment of $^{15}N$ in the $N_2$ pool (except for June 2014, see above). The incubation set-up was disassembled, and the plant was dissected into root, rhizome and leaf tissues. Pieces from each tissue were preserved for the determination of microbial $N_2$ fixation or N transfer rates (frozen at −20 °C), microbial community analyses/sequencing (frozen at −20 °C) and microscopy/single-cell analyses (see below). The remaining plant tissues were frozen at −20 °C and were later used, for example, for amino acid analyses. At least one experiment was performed for each sampling campaign.

## $N_2$ fixation and N transfer rates

The elemental and isotopic composition of 1–22 individual root, rhizome and leaf pieces from each incubated plant (for June 2014 and May 2015, also the non-incubated plant) was measured using an elemental analyser (Thermo Fisher Scientific, Flash EA, 1112 Series) coupled to a continuous-flow isotope ratio mass spectrometer (Delta Plus Advantage, Thermo Finnigan) (EA-IRMS) as described by Lehnen et al.[13]. The enrichment of $^{15}N$ in the $N_2$ pool was measured using membrane inlet mass spectrometry (MIMS, GAM200, IPI). The enrichments of $^{15}N$ in the $N_2$ pool at the beginning and at the end of the incubation were averaged (except for June 2014) for the rate calculation (June 2014: 5.4 at% $^{15}N$; May 2015: 29–46 at% $^{15}N$; April 2016: 12–17 at% $^{15}N$; August 2016: 18–32 at% $^{15}N$; May 2017: 32–40 at% $^{15}N$; September 2018: 22–32 at% $^{15}N$). Detection limits were set as a minimum change in $\delta^{15}N$ from natural abundance values within tissue types (that is, three times the s.d. of natural abundance measurements within each set of plants). This approach resulted in minimum changes in $\delta^{15}N$ values of 0.1–6.1‰ with an average of 1.8‰. When natural abundance measurements were not available (for example, failed measurements), the natural abundance values of plants and tissues closest to the same incubation conditions were used. Negative rates and rates below the detection limit were set to zero for plotting and further analysis. Root-associated microbial $N_2$ fixation rates were calculated according to Lehnen et al.[13] and are presented as µmol or nmol N fixed per gram dry weight (DW) of tissue per day (µmol $g_{DW}^{-1}$ $d^{-1}$ or nmol $g_{DW}^{-1}$ $d^{-1}$ N).

Any significant enrichment of $^{15}N$ in leaf pieces can originate only from root-associated fixation of $^{15}N_2$ and the subsequent transfer of $^{15}N$-labelled, freshly fixed N to the leaves as leaves were outside the incubation bottle with $^{15}N_2$ and rhizomes do not have a substantial role in $N_2$ fixation[13]. The measured isotopic composition of the leaf pieces was therefore used to calculate transfer rates of freshly fixed N from roots to leaves using the same rate and detection limit calculations. Transfer rates are presented as µmol or nmol N fixed per gram dry weight of tissue per day (µmol $g_{DW}^{-1}$ $d^{-1}$ or nmol $g_{DW}^{-1}$ $d^{-1}$ N).

On the basis of average rate detection limits of root-associated $N_2$ fixation rates (values obtained from propagating the minimum change through the rate equations; 0.01 µmol $g_{DW}^{-1}$ $d^{-1}$ N), plants were classified as non-$N_2$-fixing (below the average detection limit) or $N_2$-fixing (above the average detection limit) for subsequent microbial community analyses (see below).

The amount of primary production that can be sustained by root-associated $N_2$ fixation was calculated on the basis of (1) average $N_2$ fixation rates (roots and rhizomes) and N transfer rates (leaves); (2) the biomass of each tissue per incubated shoot (using an empiric conversion between dry weight and wet weight); (3) tissue-specific carbon-to-nitrogen ratios (obtained from EA-IRMS measurements); and (4) the number of shoots (counts obtained during SCUBA diving in June 2019) (Supplementary Information).

## Amino acid quantification and $^{15}N$ enrichment

During the extraction of total acid-hydrolysable amino acids and downstream processing, precautions were taken to avoid contamination by combusting all laboratory glassware before use (450 °C for 12 h). Frozen root material (0.1–1.1 g wet weight) from the $^{15}N_2$ incubations in August 2016 was freeze-dried for 2 d (Christ). Total acid-hydrolysable amino acids were extracted as follows: 20–50 mg of the freeze-dried sample were added to 3 ml of 6 M hydrochloric acid (HCl). Vials were closed with an $N_2$-flushed headspace and kept at 110 °C for 20 h. Then, 0.1–0.2 ml of the internal standard norleucine (11.1 µmol $ml^{-1}$) was added after the hydrolysis. After centrifuging the samples at 3,000 r.p.m. for 4 min, the supernatant was decanted and the pellet was dissolved in 1 ml nanopure water (MilliQ) by vortexing for 10 s and again centrifuged. The supernatant was added to the previous one, and was heated to 95 °C while flushing with $N_2$ gas until completely dried. The samples were derivatized according to a modified method by Corr et al.[48] to transform amino acids to N-acetyl i-propyl ester derivatives. In brief, amino acids were propylated with 0.63 ml of a 1:4 acetylchloride:isopropanol solution, while flushed with argon, and then kept at 100 °C for 1 h. Each vial was then cooled down to room temperature and flushed with $N_2$ until dried. Then, 0.75 ml of a derivatization solution (7.2 ml acetic anhydride, 14.4 ml triethylamine and 36 ml acetone) was added to each vial, flushed with $N_2$ while closing, vortexed and kept at 60 °C for 10 min. The samples were carefully flushed with $N_2$ until just dry. To each sample, 2 ml ethylacetate and 1 ml of saturated sodium chloride (NaCl) solution were then added and the sample was centrifuged at 2,400 r.p.m. for 3 min. The (top) organic phase was separated and carefully dried down with $N_2$. The derivatized amino acids were redissolved in 200 µl ethylacetate from which 1.5–3 µl was used for concentration and $^{15}N/^{14}N$ isotope ratio measurements. Amino acid concentrations were quantified using a gas chromatography (GC) system equipped with a flame-ionization detector (Agilent 6890N GC/7683 ALS Autosampler) and an InertCap 35 GC column (GL Sciences, 60 m × 0.32 mm × 0.50 µm). The isotope ratios ($^{15}N/^{14}N$) of individual amino acids were determined using a TRACE 1310 GC (equipped with the same column) coupled to an isotope ratio mass spectrometer (Delta V/GC IsoLink II IRMS System, Thermo Fisher Scientific).

## Nucleic acid extractions

Nucleic acid extractions of *P. oceanica* root, rhizome and leaf pieces were started by submerging several different root pieces of a plant into

liquid nitrogen and homogenizing the frozen pieces with a mortar and pestle. The powdered root material was divided into two aliquots—one aliquot was extracted using the DNeasy Plant Mini Kit (Qiagen) according to the manufacturer's instructions but excluding the RNase step. The other aliquot was extracted according to the protocol by Pjevac et al.[49]. The two extracts were pooled for each sample. The nucleic acids were then concentrated in a Speedvac (Eppendorf) at 30 °C for 50 min. The concentrate was cleaned using the Wizard DNA clean up Kit (Promega), eluted in PCR-grade water and stored at −20 °C. These nucleic acid extracts were used for Illumina-based 16S rRNA gene amplicon sequencing, Illumina-based shotgun metagenomes and for metatranscriptomes. For the PacBio-based metagenome, nucleic acids were extracted from frozen root tissue at the Max-Planck Genome Centre Cologne using the NucleoBond HMW DNA Kit (Macherey and Nagel). DNA was quality- and quantity-assessed by capillary electrophoresis (Agilent Femtopulse) and Quantus (Promega), respectively. DNA was not fragmented further and was directly used for PacBio library preparation.

Sediment for nucleic acid extractions was retrieved from cores in September 2019. Cores were sectioned and the sandy surface layer from 2–10 cm (D1) was frozen at −20 °C until further processing. DNA was subsequently extracted using the DNeasy PowerSoil Kit (Qiagen) according to the manufacturer's instructions and quantified using the Qubit dsDNA HS Assay Kit on a Qubit 2.0 Fluorometer (Invitrogen).

### 16S rRNA gene amplicon sequencing and analyses

Microbial community analyses were performed for root pieces from the $^{15}N_2$ fixation experiments to determine differences between $N_2$-fixing and non-$N_2$-fixing plants (see above). Nucleic acid extracts were sent to the Max Planck-Genome-Centre Cologne, Germany (http://mpgc.mpipz.mpg.de/home/) for barcoding PCR, library preparation and sequencing. The barcoding PCR was performed using the bacterial primers Bact341F (barcoded) and Bact805R (ref. [50]) and the DreamTaq DNA Polymerase (5 U µl$^{-1}$, Thermo Fisher Scientific). PCR started with an initial denaturation step at 98 °C for 30 s; followed by 30 cycles of 98 °C for 10 s, 55 °C for 30 s and 72 °C for 30 s; and one final elongation step at 72 °C for 5 min. The 16S rRNA gene amplicons were sequenced using the Illumina HiSeq2500 sequencing platform with 2 × 250 bp paired-end reads.

Microbial community analysis of the 16S rRNA gene amplicon data was carried out using the QIIME2 environment with a number of available plugins[51]. In brief, after importing demultiplexed reads into QIIME2, primer sequences were removed using cutadapt[52] and read pairs were joined using vsearch[53]. Error correction, trimming (to a length of 400 nucleotides) and operational taxonomic unit (OTU) clustering at the 100% similarity level was performed using deblur[54]. Taxonomy was assigned to OTUs with a sklearn-based classifier[55] through the feature-classifier plugin[56] using the full-length 16S SILVA-SSU-132 database (QIIME-compatible release from April 2018; https://www.arb-silva.de/documentation/release-132/).

As an initial assessment of whether the 16S rRNA gene amplicon datasets were representative of the sequenced root material, we calculated the ratio of bacterial to organellar reads. Some samples had a very high ratio, suggesting that the root material (and therefore also the endophytic microbial community) was not well represented. We therefore chose a cut-off of a minimum of 10% of organellar reads (out of the total reads) and, on the basis of this cut-off, three samples (all from the largest group of samples in May) were subsequently excluded from further analyses. After this initial assessment and before removing OTUs representing plastids and mitochondria, the ratio of *Celerinatantimonas*-related reads (later renamed *Ca*. C. neptuna) to organellar reads was calculated to assess whether *Ca*. C. neptuna had increased in absolute abundance in $N_2$-fixing plants relative to non-$N_2$-fixing plants[57]. After removing OTUs representing plastids and mitochondria, the final OTU table comprised 13,886 OTUs (from

a total of 31 samples that had a corresponding $N_2$ fixation rate). In a separate analysis, 16S rRNA gene amplicons sequenced from roots of a *P. oceanica* plant sampled from a meadow at the island of Pianosa were analysed equivalently.

For alpha diversity analysis, the OTU table containing the 31 samples was rarefied to a total count of 2,200 (therefore excluding 6 samples with a total OTU count <2,200; indicated in Extended Data Fig. 3) and statistical differences in alpha diversity indices between $N_2$-fixing and non-$N_2$-fixing plants were assessed with the Kruskal–Wallis pairwise test[58] using QIIME2 diversity alpha-group-significance. Beta diversity was assessed on the basis of the non-rarefied OTU table, including all 31 samples, using Aitchison principal-component analysis (PCA) through the DEICODE plugin[59] and visualized with EMPeror[60]. DEICODE also identified the OTU that contributed most to the clustering of samples in the PCA. Statistical differences in beta diversity clustering between $N_2$-fixing and non-$N_2$-fixing plants were assessed by permutational analysis of variance using QIIME2 diversity beta-group-significance testing[61]. Differential abundance testing of OTUs between $N_2$-fixing and non-$N_2$-fixing plants was performed using Songbird[62] and visualized with Qurro[63] in QIIME2. Relative abundances of bacterial OTUs were visualized with phyloseq[64].

### Metagenome sequencing and analysis

$N_2$-fixing plants (from June 2014 and August 2016) were selected for metagenome sequencing of nucleic acids extracted from root (two plants), rhizome (one plant) and leaf (one plant) tissues as well as meadow sediment (three cores). Nucleic acid extracts were sent to the Max Planck-Genome-Centre Cologne and sequencing was performed using the Illumina MiSeq platform with 2 × 250 bp paired-end reads (0.6–7.3 Gb and 8.7–10.6 Gb for plant tissue and sediment, respectively). Taxonomic assignment of raw metagenomic reads was performed using phyloFlash v3.3b3 (ref. [65]) and the parameters --tophit with the SILVA 138 database. Before analysis, reads assigned to mitochondria, chloroplasts and Eukarya were removed. Bar plots were generated using the phyloFlash_compare.pl script included in phyloFlash.

Raw metagenomic reads were later mapped onto the genome of *Ca*. C. neptuna using bbmap v.38.75 and the following parameters: minid=0.99, maxindel=1000. To reduce false positives, the rRNA operons were removed from the genome of *Ca*. C. neptuna before the mapping. For each metagenome, the number of mapped reads was normalized to the total number of reads (per million).

One of the plants with high relative abundances of *Ca*. C. neptuna (from August 2016) was also sequenced using PacBio technology (at the Max Planck Genome Centre Cologne) to obtain the genome of *Ca*. C. neptuna. In brief, the PacBio library was prepared using the SMRTbell Express Kit 2.0 (Pacific Biosciences). The library was size-selected to remove fragments smaller than 9 kb. The resulting fraction was sequenced on a single SMRT Cell (8M ZMWs) on the Sequel II system with sequencing chemistry 2.0 and binding kit 2.0 in continuous long read mode for 30 h with a total yield of 318.45 Gb (continuous long read mode). High-quality PacBio circular consensus sequencing (CCS) reads were assembled using metaFlye v.2.7 (ref. [66]). The assembly contained a circular 4.26 Mb contig with a coverage of 85, encoding 16S rRNA sequences with 100% identity to the *Celerinatantimonas*-related OTU associated with $N_2$-fixing plants and 95% identity to the 16S rRNA sequence of *C. diazotrophica*. For polishing of the *Ca*. C. neptuna metagenome-assembled genome (MAG), the 2 × 250 bp reads of the Illumina metagenome were mapped onto the metaFlye assembly with the BWA-MEM short read aligner[67] using the default settings. The resulting SAM mapping file was converted into the BAM format, sorted and indexed using SAMtools v.1.10 (ref. [68]), and subsequently used for polishing using Pilon v.1.23 (ref. [69]). The polished MAG had an estimated completeness of 100% with 0.81% contamination (CheckM (v.1.0.18)[70]) and was annotated using Prokka[71]. Mapping of CCS reads onto the *Ca*. C. neptuna-MAG was performed using minimap2 (ref. [72]).

At the time of our analysis, the genome of the closest relative *C. diazotrophica* was not available for comparison. We therefore obtained the isolate (DSM18577) from the DSMZ (Deutsche Sammlung von Mikroorganismen und Zellkulturen), grew the culture according to Cramer et al.[25] and sequenced the genome using the Illumina HiSeq 2500 platform at the MP-GC in Cologne for comparison (Extended Data Fig. 6 and Supplementary Information).

To determine the presence/absence of selected genes/pathways in the genomes of species, genera and families closely related to *Ca*. C. neptuna, we used the RAST annotation webserver[73] to annotate *Ca*. C. neptuna and 34 other genomes of other genera of the Celerinatantimonadaceae, Idiomarinaceae and Colwelliaceae. The genome accession codes and the presence/absence of selected pathways is summarized in Supplementary Data 3. Moreover, carbohydrate-active enzymes were annotated in genomes belonging to genera *Celerinatantimonas*, *Agarivorans*, *Aliagarivorans* and *Alginatibacterium* using the dbCAN meta server[74,75] using predicted protein sequences of the RAST annotation. Carbohydrate-active enzymes were predicted using HMMER, DIAMOND and Hotpep and only annotations made by ≥2 tools were retained.

## 16S rRNA phylogenetic tree reconstruction

The full-length 16S rRNA gene sequences of the closed, Prokka-annotated *Ca*. C. neptuna genome and the *Ca*. C. neptuna-related OTUs obtained from *P. oceanica* roots off the island of Pianosa, Italy (identical to those recovered at the island of Elba) were analysed phylogenetically to infer evolutionary relationships. The 16S rRNA gene sequences were added to the Silva database SSURef NR 99 release 138 (released on 11 November 2019)[76], automatically aligned using SINA[77] and the alignment was refined manually in ARB. Phylogenetic trees were calculated using distance matrix neighbour joining, maximum parsimony and maximum likelihood (FastDNAML) algorithms in ARB without position variability filters, and a consensus tree was constructed.

## Metatranscriptomic sequencing and analysis

Metatranscriptomic analyses were performed on nucleic acid extracts from June 2014. DNA was degraded using TURBO DNase (2 U μl⁻¹; Thermo Fisher Scientific), and RNA-sequencing libraries were constructed using the NEBNext Ultra II Directional RNA Library Prep Kit for Illumina (New England Biolabs). Sequencing-by-synthesis was performed on the Illumina HiSeq3000 sequencer (Illumina) with the 1 × 150 bp read mode. Library preparation and sequencing were performed by the Max Planck-Genome-Centre Cologne, Germany.

Raw transcriptomic reads were trimmed using Trimmomatic v.0.32 (MAXINFO:100:0.2, MINLEN:75)[78] after rRNA removal using SortMeRNA v.2.1 (ref. [79]) on the basis of both bacterial and archaeal rRNA databases. Non-rRNA reads were then mapped onto the genome of *Ca*. C. neptuna using Bowtie2 v.2.1.0 with the default settings[80]. Indexed BAM files were generated using samtools v.0.1.19 (ref. [68]) and transcripts per feature were quantified using featureCounts v.1.4.6 (ref. [81]) with a minimum read overlap of 75 bp (--minReadOverlap). Normalized gene transcription was subsequently quantified as transcripts per million (TPM)[82] using the formula:

$$\text{TPM}_i = \frac{c_i}{l_i} \times \frac{1}{\sum_j \frac{c_i}{l_i}} \times 10^6 \tag{1}$$

to assign each feature $i$ a TPM value where $c$ is the feature count, $l$ is the length in kilobases and $j$ is all features. TPM values were visualized together with the mapped Illumina short reads, the mapped PacBio CCS reads in circular genome figures using BRIG[83]. Furthermore, TPM values were normalized to the average TPM of a set of housekeeping genes (*rpoA*, *rpoB*, *ftsZ*, *rho*, *recN*, *gyrB*, *recA* and *gyrA*)[84–89] for each of the five samples from June 2014 (one sample did not return enough *Ca*. C. neptuna reads to be mapped) to visualize the selected pathways (Extended Data Fig. 7).

## Fixation, embedding and sectioning of *P. oceanica* root material

Root pieces were preserved at the end of the ¹⁵N₂ tracer incubations (see above) in paraformaldehyde solution (4% (w/v) final concentration in filtered seawater) at ambient/room temperature for 1 h. Root pieces were then washed with phosphate-buffered saline (PBS) solution and in nanopure water (MilliQ) for 15 min and 10 min, respectively. The pieces were then dehydrated in 96% ethanol for 2 min and air dried for 30 min. The fixed root material was stored at −20 °C until further processing.

Before resin infiltration, the formaldehyde-fixed root material was dehydrated using an ethanol series of 30%, 50%, 70%, 80% and 90% (once), and 100% (twice) for 10 min each. Pieces were then infiltrated with resin by stepwise increases of London Resin White (LRW; Sigma-Aldrich) with concentrations of 25%, 50% and 75% (each once), and 100% (twice) LRW in ethanol (modified from McDonald[90]). Each infiltration step was performed for 15 min, and the root pieces were then centrifuged for 5–10 min using a benchtop centrifuge. For polymerization of the resin, the individual root pieces were submerged in 100% LRW resin inside gelatin capsules or Eppendorf tubes. Capsules or tubes were placed inside a gas-tight bag, which was flushed with N₂ gas for 1 h. The gas-tight bags were subsequently kept at 65 °C for 4–5 d. Semi-thin (thickness, 0.5–1 μm) sections of root pieces were cut with glass knives using the Leica UC7 Ultramicrotome (Leica Microsystems). The sections were placed onto Polysine Adhesion Slides (Thermo Fisher Scientific) or indium tin oxide (Präzisions Glas & Optik) glass slides and were dried on a heating plate at 60 °C for 5 min. The semi-thin sections were stored at 4 °C until further processing.

## FISH analysis of semi-thin root sections

To visualize *Ca*. C. neptuna cells in *P. oceanica* roots, we designed four FISH-probes targeting the 16S rRNA gene of *Celerinatantimonas* spp. (that is, the 16S rRNA genes of the MAG, *C. diazotrophica* and *C. yamalensis*). The probe set had some matches outside *Celerinatantimonas*, which were, however, not present in our 16S rRNA gene amplicon dataset. All individual FISH probes, the probe set (all four probes together) and the EUB338-I (positive control) and the NON338 (negative control) probes[91,92] were used to determine melting curves using the *C. diazotrophica* DSM18577 culture. All four probes (5′−3′: Cel_442 (ACCCT TCCTCACAAC), Cel_186 (TCCCCTGCTTTGGTCCGTAG), Cel_660 (AAATTCTACCTCCCTCTACA) and Cel_227 (TAATCTCACTTGGGT GCATC)) were then used in combination for all hybridizations using a formamide concentration of 25%. All FISH probes were obtained from Biomers and were labelled with two Atto550 molecules[93]. Hybridizations were performed on semi-thin root sections (see above) using standard hybridization protocols with the following modifications. Root sections were encircled with a water-repellent barrier oil layer using an oil PAP PEN (G. Kisker) to ensure that the sections were always submerged in the respective solutions. Hybridization using the *Celerinatantimonas* probe mix was performed at 35 °C for 2 h (hybridization solution with 0.9 M NaCl, 0.02 M Tris-HCl, 25% formamide and 0.01% (w/v) SDS). After hybridization, thin sections were sequentially washed in washing buffer (0.01% (w/v) SDS, 20 mM Tris-HCl, 5 mM EDTA and 149 μM NaCl) for 45 min at 37 °C, in 4 °C cold PBS for 15 min and in nanopure water (MilliQ) for 5 min. Sections were placed in ethanol (96%) for 2 min and then air-dried. DNA was counterstained with 4′,6-diamidino-2-phenylindole (DAPI). The root sections were covered with a mixture of Citifluor AF1 (Citifluor) and Vectashield (Vector Laboratories) (ratio of 1:4) and a coverslip for microscopy. FISH was performed on sections prepared from root material from June 2014, April 2016 and August 2016 for cell counts, quantitative visualization and images. Additional material was sampled and preserved in June 2019 and processed for FISH imaging.

## Microscopy and nanoSIMS analysis

Bacterial cells hybridized with the *Celerinatantimonas* FISH probe (that is, *Ca*. C. neptuna cells) were counted manually using epifluorescence

microscopy on several root cross-sections (thickness, 0.5–1 μm) from April 2016 (non-$N_2$-fixing), June 2014 ($N_2$-fixing) and August 2016 ($N_2$-fixing). Individual representative images were taken from these root cross-sections for illustration. Additional FISH images were obtained from root material collected in June 2019.

To obtain a proxy for the contribution of *Ca*. C. neptuna to total biomass within the complete root cross-sections, the distribution of *Ca*. C. neptuna cells was mapped using epifluorescence images acquired using the Zeiss Axio Imager. An M2 microscope at 100-fold magnification equipped with an automated XYZ stage (Märzhäuser Wetzlar, SCAN IM, 130 × 85, 2 mm). Approximately 10 × 10 images in horizontal directions and 30 images in the vertical direction were taken to cover a full cross-section. Each image was composed of three channels: DAPI (blue), autofluorescence (green) and FISH (red/orange). The raw image stack was processed using the Zeiss microscope software ZEN (ZEN 3.2 blue edition). In brief, images were first stitched and corrected for shading. A deconvolution algorithm and orthogonal projection was applied to correct for noise and light scattering into the image from planes above and below the focal plane. The channels were then merged to optimize the contrast of positively hybridized cells attached to the root tissue. In the resulting RGB image, overlapping signals of DAPI and FISH appear in pink and autofluorescence in green. Subsequently, the images were processed in MATLAB (Mathworks 2018b) to determine the area occupied by FISH-positive cells and root tissue. The area surrounding the root was masked, and the image was decomposed into the red, green and blue channels. Root tissue was determined based on the green channel, which was binarized with a threshold of 1% of its maximum intensity. In the binarized image, root tissue appears white (1) while the remaining pixels are black (0). To determine the total root area, all white pixels were integrated (1 px represents 0.1 μm). For the positively hybridized cells, a similar procedure was applied on the basis of the combined red and blue channel. However, the root tissue, namely the rhizoplane, epidermis, hypodermis and the innermost areas of the stele, were masked due to strong autofluorescence signals along the whole spectrum and were excluded from the processing of the FISH-positive cellular area. Manual cell counts had confirmed that these root tissues did not contain any *Ca*. C. neptuna cells, and the exclusion of these areas therefore did not bias our automated analyses. Areas of root tissue and *Ca*. C. neptuna cells were later used in the nanoSIMS-based mass balance assuming that the occupied area is representative of the biomass contribution. This quantification was performed on semi-thin root sections from April 2016 (non-$N_2$-fixing), June 2014 ($N_2$-fixing) and August 2016 ($N_2$-fixing).

For visualization purposes (Extended Data Fig. 4), the black and white image of the root image was inverted, such that the root tissue appears in black. The root image was then overlain with the positively hybridized cells in red (Extended Data Fig. 4). To better visualize the location of cells in the root cross-section, cells were artificially blurred by applying a Gauss filter at increasing kernel sizes (3–20 pixel).

After microscopy, the Citifluor–Vectashield-mix was washed off the sections with nanopure water (MilliQ) three times, and the sections were air-dried on their slides. Before nanoSIMS measurements, the root sections were sputter-coated with 10 nm gold (Au) using the Leica EM ACE600 (Leica Microsystems) sputter coater to ensure conductivity. For nanoSIMS measurements, the area of interest was presputtered for 2 min with a positively charged caesium ($Cs^+$) primary ion beam to implant $Cs^+$ on the sample surface. Sample surfaces were rastered with a $Cs^+$ primary ion beam with a current of 1.5 pA. Primary ions were focused into a nominal ≤100 nm spot diameter. The image resolution was 256 px × 256 px with a dwelling time of 1 ms per pixel. Analysed areas were 20 μm × 20 μm. Secondary ion counts of carbon ($^{12}C^-$), nitrogen (as $^{12}C^{14}N^-$ and $^{12}C^{15}N^-$), phosphorus ($^{31}P^-$) and sulfur ($^{32}S^-$) were recorded simultaneously by the electron multiplier detectors of the multicollection system of the instrument.

To have a better statistical representation of the $^{15}N$ enrichment in root tissue and *Ca*. C. neptuna cells, a total of 167 NanoSIMS images (37 images for June 2014 section and 130 images for August 2016) were processed using a semi-automated algorithm. First, all nanoSIMS images were processed using look@nanoSIMS[94]. All planes (40 for each image) were drift-corrected and accumulated. NanoSIMS ($^{12}C^{14}N^-$) images were manually aligned and overlapped with their corresponding epifluorescence microscopic images. Next, the NanoSIMS and epifluorescence microscopic images were exported and further processed using a custom-developed MATLAB algorithm. On the basis of the epifluorescence images, binary matrices for positively hybridized cells and root material were determined as described above. These binary matrices were pixel-wise multiplied with the $^{15}N/^{14}N$ isotope ratio matrices, yielding an isotope ratio matrix for *Ca*. C. neptuna cells and an isotope ratio matrix for root tissue. The isotope ratios within the two matrices were then averaged yielding one isotope ratio value for *Ca*. C. neptuna cells and one for root tissue per nanoSIMS image. Isotope ratios ($r = {}^{15}N/{}^{14}N$ from $^{12}C^{15}N^-/^{12}C^{14}N^-$) were then converted to atomic percentage using $^{15}N$ at% $= r/(r + 1) × 100$ (at%). Of all 167 images, 78 images were of root tissue only whereas 89 contained both root tissue and *Ca*. C. neptuna cells.

To account for the different labelling percentages of the $N_2$ pool in June 2014 and August 2016, relative incorporation (per day) was calculated using the $^{15}N$ at% (at%$_{cell}$) of (1) *Ca*. C. neptuna cells; (2) root tissue with *Ca*. C. neptuna cells close by (that is, nanoSIMS image with root tissue and *Ca*. C. neptuna cells present) as well as root tissue alone (that is, nanoSIMS images without *Ca*. C. neptuna cells present), natural abundance background $^{15}N$ (June 2014: 0.367194; and August 2016: 0.367456; at%$_{NA}$) from the bulk biomass, the enrichment of $^{15}N$ in the $N_2$ pool (at%$_{N2}$, measured by MIMS; see above) and incubation time ($t$) as follows:

$$\text{Relative incorporation(\%)} = (\text{at\%}_{cell} - \text{at\%}_{NA})/(\text{at\%}_{N2} - \text{at\%}_{NA}) × 1/t × 100 \quad (2)$$

As the application of FISH procedures can lead to underestimates of the isotopic ratio[95], the calculated relative incorporation represents a minimum estimate.

Finally, to visualize a larger area of the root cross-section (Fig. 3d, e), 11 nanoSIMS images that overlapped by 5 px were stitched based on the position of the nanoSIMS XYZ stage. Due to 3D effects, the raster areas were not always aligned perfectly. Thus, a custom developed cross-correlation algorithm (MATLAB, Mathworks 2018b) was applied with a maximum allowed shift of 5 px to improve stitching. In case offsets were larger, the position was manually corrected. Look@nanosims was modified to allow for reading of the stitched images and subsequent processing according to the same procedure as described above. As nanoSIMS measurements were performed on root cross-sections in the embedding medium, regions without plant material had low count-statistics, which can lead to false $^{15}N/^{14}N$ ratios. To not overemphasize these regions with low count-statistics, we used the thresholding method implemented in look@nanoSIMS for Fig. 3e (for comparison, an example raw image is presented in Extended Data Fig. 4d). Custom codes for processing of nanoSIMS data are available (see the 'Code availability' section).

## STEM analysis

Paraformaldehyde-fixed root pieces (from June 2014) that were previously used for FISH (see above) were also used for scanning transmission electron microscopy (STEM) imaging. Thin sections (~70 nm) were prepared with the Leica UC7 Ultramicrotome (Ultracut UC7, Leica Microsystems) using a diamond knife, mounted on formvar-coated slot-grids (Agar Scientific). Sections were stained with osmium tetroxide ($OsO_4$), followed by 0.5% aqueous uranyl acetate (Science Services) for 20 min and 2% Reynold's lead citrate for 6 min, with three washing steps between each step. Sections were imaged at 20–30 kV using the Quanta 250 FEG scanning electron microscope (FEI) equipped with a STEM detector using the xT microscope control software (v.6.2.6).

## Statistics and reproducibility

No statistical methods were used to predetermine sample size and experiments were not randomized. The investigators were not blinded to allocation during experiments and outcome assessment.

For Fig. 3a–c, respectively, the fluorescence images are representative of $n = 3$ images from 1 sample; $n = 9$ images from 4 sections of 1 sample; and $n = 23$ images from 4 sections of 1 sample. In Fig. 3d, e, the stitched images are representative of $n = 2$ images from 2 samples.

In Extended Data Fig. 4d, the correlative images (FISH and nanoSIMS) are representative of $n = 167$ measurements from 2 samples; 11 of the 167 images were merged and illustrated in Fig. 3d, e. Of the 167 measurements, 89 measurements contained both root tissue and *Ca*. C. neptuna cells while 78 measurements contained root tissue only.

In Extended Data Fig. 5a, b, the STEM images are representative of $n = 10$ images from 1 sample. In Extended Data Fig. 5c, the epifluorescence image is representative of $n = 11$ from 4 sections of 1 sample.

## Reporting summary

Further information on research design is available in the Nature Research Reporting Summary linked to this paper.

## Data availability

Raw reads of the 16S rRNA gene amplicon sequencing, the MAGs of *Ca*. C. neptuna and *C. diazotrophica* (DSM18577), and the mapped reads of the transcriptomes are available under Bioproject number PRJEB37438 at the European Nucleotide Archive (ENA). Sequences that were included in the phylogenetic tree are available in Supplementary Data 1 (with accession numbers and references) and as a tree file (Supplementary Data 2). The comparison of 34 genomes for presence/absence of specific genes and/or pathways is available in Supplementary Data 3 including their accession numbers. The PhyloFlash results (as presented in Extended Data Fig. 2) are available in Supplementary Data 4. Publicly available sequences used for phylogenetic tree construction and genome comparison can be found under their respective accession numbers at NCBI (https://www.ncbi.nlm.nih.gov/) or ENA (https://www.ebi.ac.uk/ena/browser/home). Ribosomal subunit databases used for taxonomic classification can be found at the SILVA rRNA database (https://www.arb-silva.de/). Source data are provided with this paper.

## Code availability

MATLAB codes used for processing of eddy correlation data and nanoSIMS data can be found at GitHub (https://github.com/SoerenAhmerkamp/EddyCorrelation and https://github.com/SoerenAhmerkamp/NanoSIMS/).

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

**Acknowledgements** We thank the staff at the HYDRA Marine Sciences and HYDRA Fieldwork for field sampling and assistance as well as scientific discussions during our study; staff at the National Park Tuscan Archipelago, Portoferraio, Italy for granting access to the protected waters of the Island of Pianosa (permit no. 2930/2017); N. Dubilier for valuable discussions and the organization of two workshops on symbiotic ecology at the Elba Field Station; J.-H. Hehemann, B. Reinhold-Hurek, T. Hurek, L. van Niftrik, M. Marín and B. Kartal for discussions; and T. Alarcon Schumacher, P. Bourceau, C. Cornet, J. Dekaezemacker, P. Downes, A. Frayssinet, B. Fuchs, P. Hach, K. Imhoff, F. Moin Jalaluddin, A. Kidane, K. Kitzinger, G. Klockgether, S. Lilienthal, M. Maeke, S. Murugan, W. Neweshy, S. Piosek, S. Robert, N. Rujanski, S. Schorn, A. Schwedt, P. Stücheli, D. Tienken and B. Vekeman for technical assistance and help with sampling and sample preparations. C.J.S. was funded by internal Eawag funds. This study was funded by the Max Planck Society.

**Author contributions** W.M., N. Lehnen, H.K.M. and M.M.M.K. performed $N_2$ fixation experiments, and W.M. and N. Lehnen analysed $N_2$ fixation data. N. Lehnen carried out embedding, FISH and microscopy on root cross-sections as well as nucleic acid and amino acid extractions of root pieces. S.A. performed EC measurements and analysed and processed microscopic and nanoSIMS images. N. Lehnen, J.S.G. and B.T. analysed amplicon-sequencing data, metagenomes, MAGs and metatranscriptomes, and performed genome comparisons. S.L. carried out nanoSIMS measurements. P.Y. and H.G.-V. carried out initial bioinformatics analyses. C.J.S. provided amino acid data. M.W. and C.L. carried out field sampling and in situ field measurements. N. Leisch carried out STEM and provided help with embedding, sectioning and FISH. M.M.M.K. conceived the project. W.M., N. Lehnen, H.K.M., J.M. and M.M.M.K. designed the study. W.M., J.M. and M.M.M.K. wrote the manuscript with contributions from all of the co-authors.

**Funding** Open access funding provided by Max Planck Society.

**Competing interests** The authors declare no competing interests.

**Additional information**
**Correspondence and requests for materials** should be addressed to Wiebke Mohr.

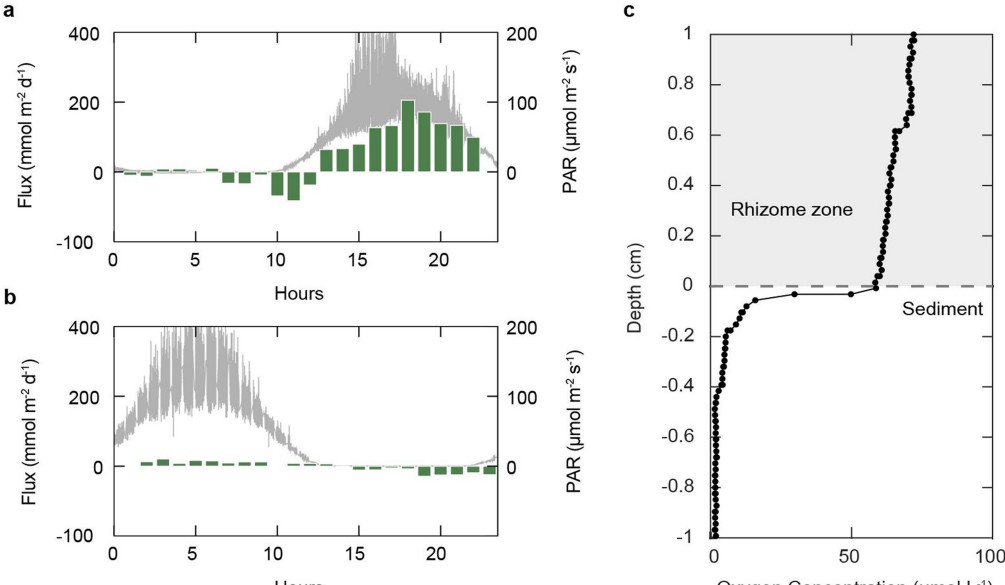

**Extended Data Fig. 1 | Productivity in seagrass sediments and non-vegetated sediments.** Hourly-averaged oxygen fluxes (green) and photosynthetically active radiation (PAR; grey) measured over a daily cycle in June 2019 for the *P. oceanica* meadow (**a**) and unvegetated sandy sediments (**b**). Negative fluxes during night indicate net respiration, whereas positive fluxes during the day indicate net photosynthesis. The small positive oxygen fluxes in the lower panel indicate the presence of benthic algae. (**c**) Oxygen ($O_2$) concentrations at the rhizome zone-sediment interface showing the depletion of $O_2$ at about 4 mm depth.

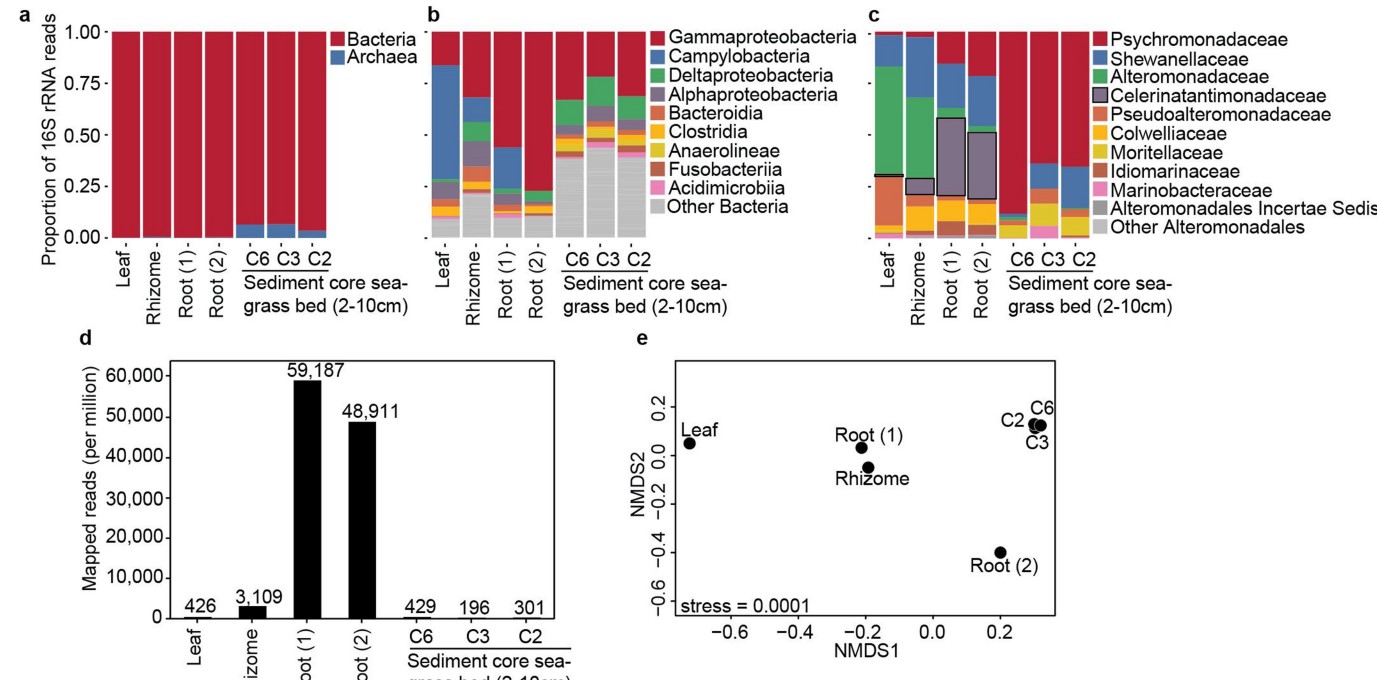

**Extended Data Fig. 2 | Composition of microbial communities associated with *P. oceanica* plant tissues and seagrass sediment.** ((**a**)–(**c**)) 16S rRNA gene-based community composition of the top ten taxa on domain-level (**a**), class-level (only Bacteria) (**b**), and families within the order Alteromonadales (**c**) in metagenomes associated with *P. oceanica* plant tissues (Leaf, Rhizome, Root 1, Root 2) and the sediment within the *P. oceanica* seagrass meadow (replicate cores C2, C3, C6; 2-10 cm depth horizon). Note that while reads associated with Alteromonadales were detected in all samples, reads assigned to Celerinatantimonadaceae were only detected in plant-associated samples (mainly roots and rhizome). Reads taxonomically assigned to either mitochondria, chloroplasts or Eukarya were not included in the analysis. (**d**) Read recruitment to *Ca*. C. neptuna from metagenomes sampled from

different *P. oceanica* plant tissues and seagrass sediment. Metagenomic reads mapped onto the genome of *Ca*. C. neptuna and the number of mapped reads normalized to one million total reads is shown above each bar. Reads mapping to rRNA genes were not counted in this analysis. (**e**) Non-metric multidimensional scaling (NMDS) ordination plot showing changes in the prokaryotic community composition (Class-level) associated with different *P. oceanica* plant tissues and seagrass bed sediment. Community composition on Class-level (see panel b) was derived from metagenomic 16S rRNA gene sequences sampled from *P. oceanica* plant tissues (Leaf, Rhizome, Root 1, Root 2) and the sediment within the *P. oceanica* seagrass meadow (replicate cores C2, C3, C6; 2-10 cm depth horizon). Ordinations are based on Bray-Curtis dissimilarity.

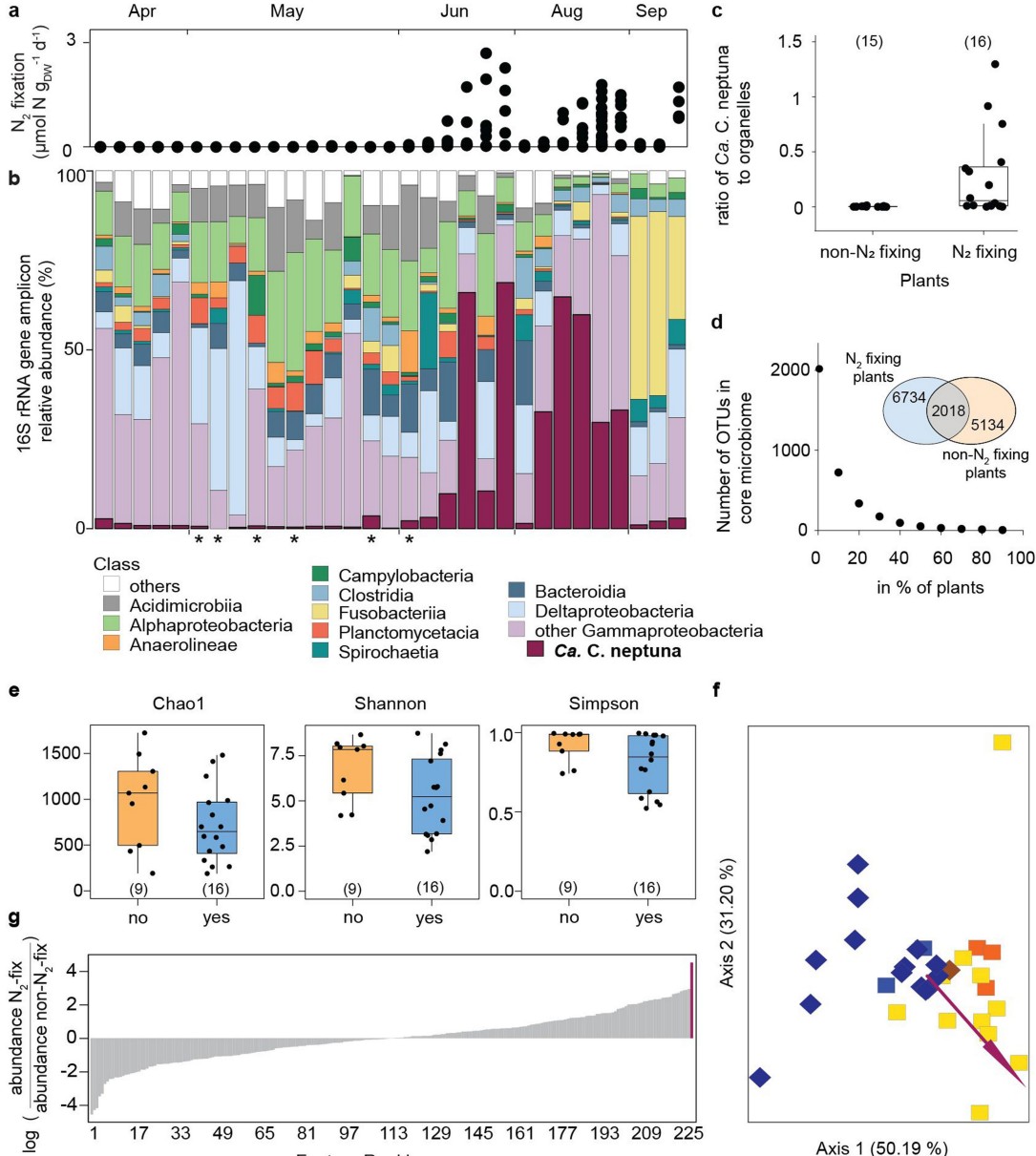

**Extended Data Fig. 3 | Root-associated microbial community analyses of N₂-fixing and non-N₂-fixing *P. oceanica* plants.** (**a**) $N_2$ fixation rates of roots of individual plants with each symbol representing individual root pieces. (**b**) Relative abundance of 16S rRNA gene-based OTUs ('others' are classes with less than 1.5%) in individually analysed plants. Asterisks indicate plants with sequencing results below the threshold for the calculation of α-diversity. (**c**) Ratio of *Celerinatantimonas*–related reads to organellar reads. Lines, boxes, and error bars represent mean, 25th and 75th percentiles and standard deviation, respectively. (**d**) Number of shared OTUs as a function of their presence in an increasing number of plants (as % of plants in both categories) with Venn diagram showing the number of unique and shared OTUs. (**e**) α-diversity indices where boxes indicate second and third quartiles, whiskers indicate first and fourth quartiles, lines indicate median values and black dots are the individual plants. Differences in α-diversity indices were not statistically significant (Kruskal-Wallis pairwise tests). (**f**) First two axes of a principal component analysis (PCA) using Aitchison distance based on OTU counts, highlighting statistically significant microbial community compositions in $N_2$-fixing (square symbols) and non-$N_2$-fixing (diamond symbols) plants (pairwise Permanova, peudo-F = 7.9, q-value = 0.001). Colors indicate seasons (spring (blue), summer (yellow) and autumn (orange); the brown diamond represents a spring and a summer sample on top of each other). Arrow indicates the OTU contributing most to the clustering of samples, i.e. *Ca.* C. neptuna. (**g**) Ranking of differentially abundant OTUs (x-axis). Negative and positive log ratios indicate higher abundance in non-$N_2$-fixing and $N_2$-fixing plants, respectively. Highlighted in magenta is the *Ca.* C. neptuna-OTU, the highest ranked OTU positively associated with $N_2$ fixation. The number of plants (n) included in the analyses is indicated in parentheses in panels c and e.

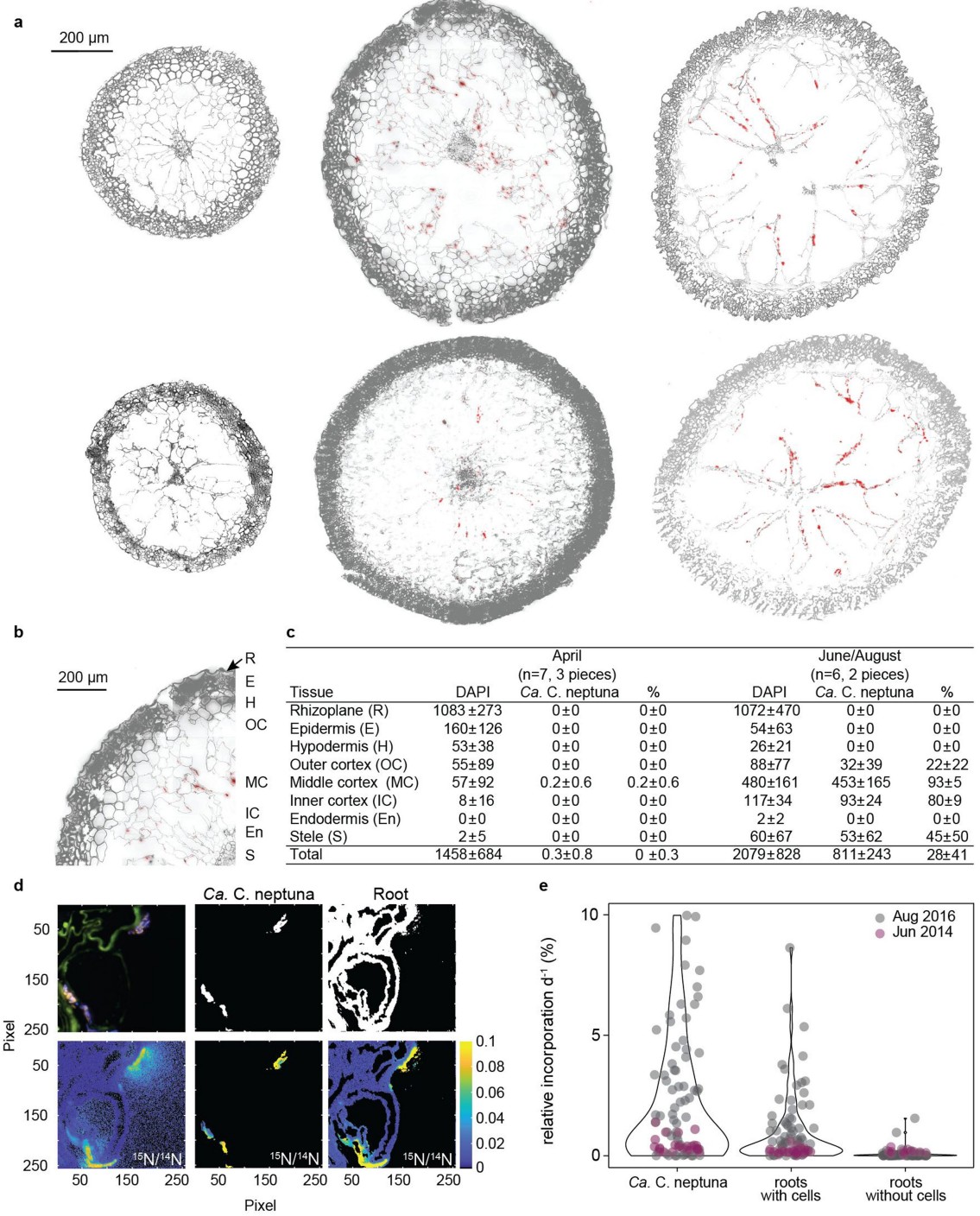

| | April (n=7, 3 pieces) | | | June/August (n=6, 2 pieces) | | |
|---|---|---|---|---|---|---|
| Tissue | DAPI | Ca. C. neptuna | % | DAPI | Ca. C. neptuna | % |
| Rhizoplane (R) | 1083 ±273 | 0±0 | 0±0 | 1072±470 | 0±0 | 0±0 |
| Epidermis (E) | 160±126 | 0±0 | 0±0 | 54±63 | 0±0 | 0±0 |
| Hypodermis (H) | 53±38 | 0±0 | 0±0 | 26±21 | 0±0 | 0±0 |
| Outer cortex (OC) | 55±89 | 0±0 | 0±0 | 88±77 | 32±39 | 22±22 |
| Middle cortex (MC) | 57±92 | 0.2±0.6 | 0.2±0.6 | 480±161 | 453±165 | 93±5 |
| Inner cortex (IC) | 8±16 | 0±0 | 0±0 | 117±34 | 93±24 | 80±9 |
| Endodermis (En) | 0±0 | 0±0 | 0±0 | 2±2 | 0±0 | 0±0 |
| Stele (S) | 2±5 | 0±0 | 0±0 | 60±67 | 53±62 | 45±50 |
| Total | 1458±684 | 0.3±0.8 | 0 ±0.3 | 2079±828 | 811±243 | 28±41 |

**Extended Data Fig. 4 | Distribution, abundance and activity of Ca. C. neptuna.** (**a**) Stitched epifluorescence images (black-and-white inverted) of root cross-sections from April 2016 (left), June 2014 (middle) and August 2016 (right) showing primary locations of Ca. C. neptuna cells (as visualized by FISH) in red (signal-amplified for easier visualization; Methods), (**b**) Epifluorescence image (from panel a) indicating the different root tissues (as in c), (**c**) absolute and relative abundance of Ca. C. neptuna cells (visualized by FISH) in the different root tissues of individual P. oceanica root sections (0.5 – 1.0 µm thickness), (**d**) Correlative epifluorescence (FISH and autofluorescence) and nanoSIMS ($^{15}N/^{14}N$) images (left) with the retrieved surface areas and correlative nanoSIMS data for Ca. C. neptuna cells (middle) and root tissue (right) (Methods), x- and y-axes are in pixels with 256 pixels = 20 µm, and the color scale represents the $^{15}N/^{14}N$ ratio; (**e**) single-cell $^{15}N_2$ fixation activity (Ca. C. neptuna cells) or transfer of freshly fixed N (root tissue) as relative incorporation per day in Ca. C. neptuna cells, root tissue with Ca. C. neptuna cells in the same image (roots with cells) and root tissue without Ca. C. neptuna cells in the same image (roots without cells) obtained from a total of 167 measurements as shown in (**d**) from two cross-sections.

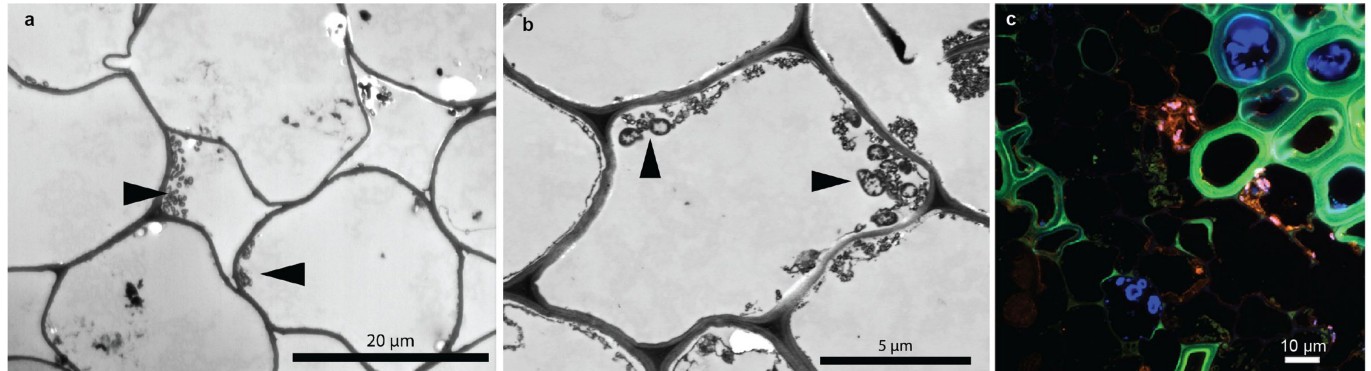

**Extended Data Fig. 5 | STEM and epifluorescence images of seagrass root sections. (a, b)** STEM images with black arrowheads indicating bacteria residing in intercellular space (**a**) as well as inside plant cells (**b**). Based on the location as well as the relative and absolute abundances of *Ca*. C. neptuna in root cross-sections (Extended Data Fig. 4), the majority of bacteria visible are likely *Ca*. C. neptuna cells. (**c**) Epifluorescence image of a root cross-section showing *Ca*. C. neptuna cells (overlay image of DAPI (blue), autofluorescence (green/orange) and FISH-positive cells (pinkish color due to the overlap of DAPI and FISH probe (orange) signals) inside the stele.

| | Size (Mbp) | Contigs | GC (%) | rRNA operons | tRNAs | ORFs | ORFs (hypoth.) | 16S rRNA | ANI (%) | AAI (%) |
|---|---|---|---|---|---|---|---|---|---|---|
| *Ca.* C. neptuna | 4.26 | 1 | 42.9 | 6 | 75 | 3727 | 1382 | n/a | n/a | n/a |
| *C. diazotrophica* (DSM18577) | 4.01 | 1 | 43.7 | 6 | 79 | 3600 | 1201 | 95.0 | 78 | 69 |

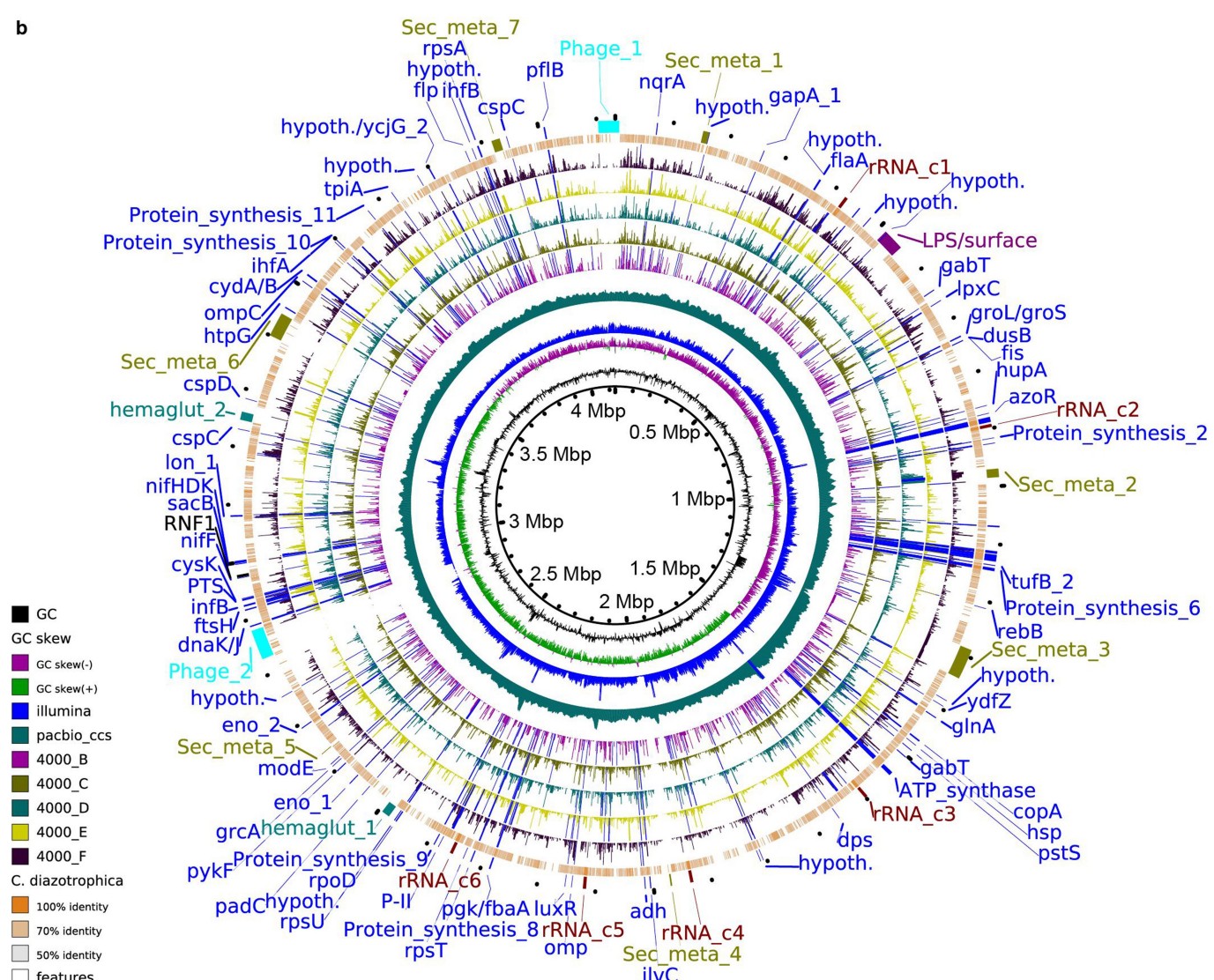

**Extended Data Fig. 6 | Genome and transcriptomes of *Ca*. C. neptuna.**
General genome characteristics for the novel species *Ca*. C. neptuna and its closest relative *C. diazotrophica* (DSM18577) (top). *Ca*. C. neptuna genome is represented circularized (bottom). Circles from the inside to the outside: (1) GC content, (2) GC skew, (3) Illumina metagenome (June 2014) coverage (coverage range 0-250), (4) PacBio metagenome (August 2016) coverage (coverage range 0-200), (5-9) gene transcription in Illumina-based transcriptomes plotted as TPM for protein coding locus tags (TPM range 0-1000; genes with TPM values > 1000 are highlighted in dark blue), (10) regions present in *C. diazotrophica* based on BLASTn searches, shown are regions with similarity between 50% (grey) – 100% (orange), (11) selected features (genes and gene clusters) including highly transcribed ones (blue, mean expression across all transcriptomes >1000 TPM). A list of genes and features is included in the Supplementary Information.

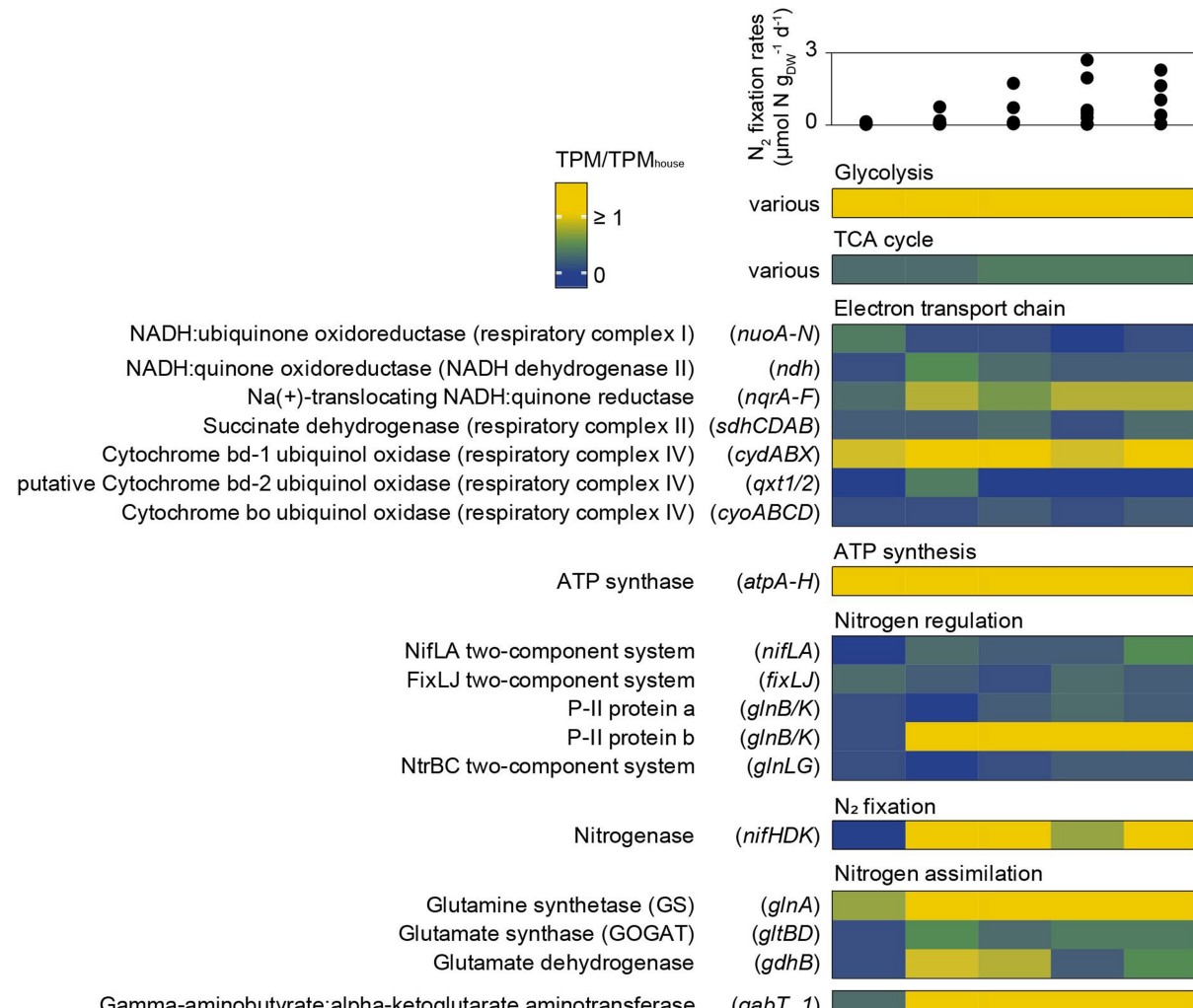

**Extended Data Fig. 7 | Gene transcription of selected genes/pathways.**
Normalized gene transcription (TPM/TPM$_{house}$ = TPM relative to TPM of housekeeping genes) of selected *Ca*. C. neptuna pathways in transcriptomes of five N$_2$-fixing plants (1 plant = 1 column). The genes averaged for analysis are indicated in parentheses. For glycolysis and the TCA cycle, the following genes were pooled for analysis: *pgi, pfkA, fbaA, tpiA, gapA, pgk, gpmA, gpmI, eno, pykF*

and *gltA, acnB, icd, lpdA, sucA, sucB, sucC, sucD, sdhC, sdhD, sdhA, sdhB, fumB, mdh, mqo*, respectively. Measured N$_2$ fixation rates of the respective plants are shown on top. The (non-linear) color-coding was chosen so that transcription similar to housekeeping genes or higher is presented as yellow while lower expression levels are gradient-colored between blue and yellow.

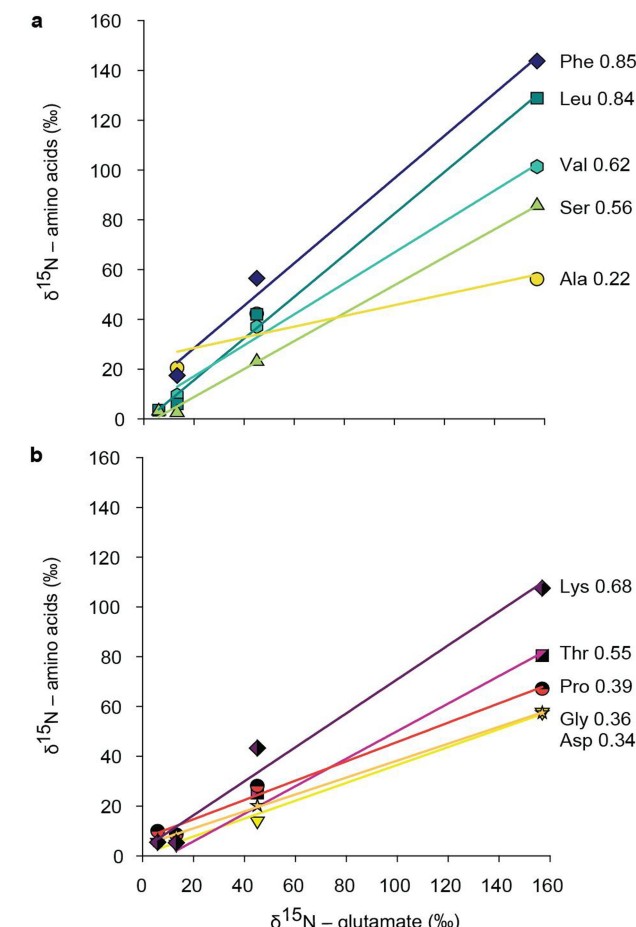

**Extended Data Fig. 8 | Enrichment of $^{15}$N in amino acids.** Enrichment of $^{15}$N in individual amino acids relative to the enrichment of $^{15}$N in glutamate, the first amino acid synthesized upon $N_2$ fixation, in root-extracted proteins after incubations of plants with $^{15}N_2$ (Methods; up to four plants that yielded sufficient amino acids for $\delta^{15}$N measurement are shown here). **a**) Amino acids synthesized from intermediates of the glycolysis or the pentose phosphate pathway and **b**) amino acids synthesized from intermediates of the TCA cycle or via transamination of another amino acid (or intermediates) with glutamate. Slopes of the linear regressions are given behind the three-letter code for each amino acid. A slope of 1.0 indicates $^{15}$N enrichments that are identical to those of glutamate with smaller slopes indicating lower $^{15}$N enrichments (see Supplementary Discussion). Phe, phenylalanine ($R^2$=0.991); Leu, Leucine ($R^2$=0.995); Val, Valine ($R^2$=0.993); Ser, Serine ($R^2$ = 0.998); Ala, Alanine ($R^2$ = 0.819); Lys, Lysine ($R^2$ = 0.979); Thr, Threonine ($R^2$ = 0.986); Pro, Proline ($R^2$ = 0.988); Gly, Glycine ($R^2$ = 0.992); Asp, Aspartate ($R^2$ = 0.999).

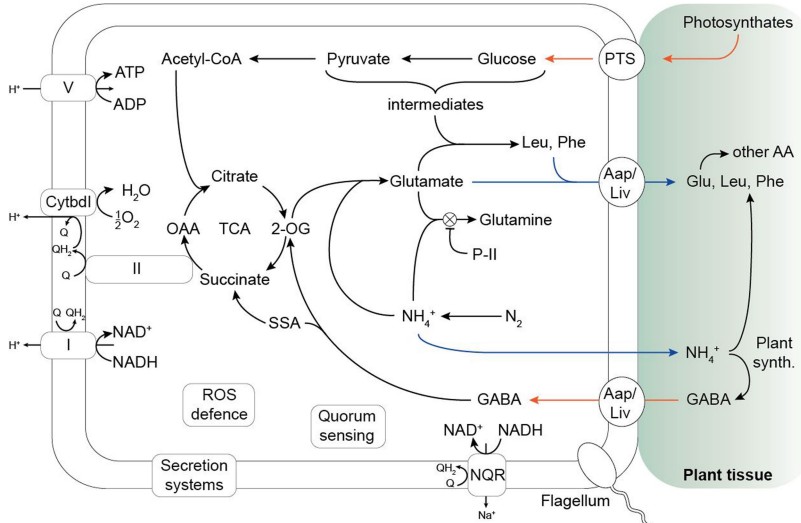

**Extended Data Fig. 9 | Model of main metabolic interactions of *Ca*. C. neptuna and *P. oceanica*.** Blue and orange arrows indicate transfer of metabolites from *Ca*. C. neptuna to *P. oceanica* and vice versa, respectively. I, respiratory complex I (NADH dehydrogenase); II, respiratory complex II (succinate dehydrogenase); V, respiratory complex V (ATP synthase); CytbdI, *bdI*-type terminal quinol oxidase; OAA, oxaloacetate; TCA, tricarboxylic acid cycle; 2-OG, 2-oxoglutarate; SSA, succinic semialdehyde; PTS, phosphotransferase system (sugar transport); Aap/Liv, Amino acid permease / branched-chain amino acid transporter; GABA, 4-aminobutyrate; NQR, sodium-translocating NADH:ubiquinone reductase; Glu, glutamate; Leu, leucine; Phe, phenylalanine; AA, amino acids.

**Extended Data Table 1 | Nutrient concentrations associated with *P. oceanica* meadow and nearby sediments**

| | Water column | | | |
|---|---|---|---|---|
| **a** | May 2017 | Jun 2014 | Jun 2019 | Sep 2018 |
| Phosphate ($\mu$mol L$^{-1}$) | b.d. (3) | b.d. (2) | b.d. (2) | b.d. (32) |
| Silicate ($\mu$mol L$^{-1}$) | 1.0 ± 0.1 (3) | 0.2 ± 0.1 (2) | 0.5 ± 0.1 (2) | 0.2 ± 0.2 (32) |
| Nitrite ($\mu$mol L$^{-1}$) | b.d. (3) | b.d. (2) | b.d. (2) | b.d. (32) |
| Nitrate ($\mu$mol L$^{-1}$) | 0.4 ± 0.1 (3) | b.d. (2) | b.d. (2) | b.d. (32) |
| $NO_x^-$ ($\mu$mol L$^{-1}$) | 0.4 ± 0.1 (3) | b.d. (2) | b.d. (2) | b.d. (32) |
| Ammonium ($\mu$mol L$^{-1}$) | 0.4 ± 0.1 (3) | n.a. | b.d. (2) | b.d. (3) |

| | Sediment (June 2019) | |
|---|---|---|
| **b** | Seagrass sediment | Non-vegetated sediment |
| $NO_x^-$ ($\mu$mol L$^{-1}$) | 0.3 (2) | b.d. (1) |
| Ammonium ($\mu$mol L$^{-1}$) | 3.6 (2) | 1.7 (1) |

Nutrient concentrations from water column collected above or next to seagrass meadow (a) and porewater (average of upper 20 cm) of seagrass sediments and non-vegetated sediments (b). Numbers in brackets indicate the number of samples measured for each campaign. b.d. = below detection limit, n.a. = not available.

| | |
|---|---|

# Reporting Summary

## Statistics

For all statistical analyses, confirm that the following items are present in the figure legend, table legend, main text, or Methods section.

| n/a | Confirmed | |
|---|---|---|
| ☐ | ☒ | The exact sample size (*n*) for each experimental group/condition, given as a discrete number and unit of measurement |
| ☐ | ☒ | A statement on whether measurements were taken from distinct samples or whether the same sample was measured repeatedly |
| ☐ | ☒ | The statistical test(s) used AND whether they are one- or two-sided *Only common tests should be described solely by name; describe more complex techniques in the Methods section.* |
| ☒ | ☐ | A description of all covariates tested |
| ☐ | ☒ | A description of any assumptions or corrections, such as tests of normality and adjustment for multiple comparisons |
| ☐ | ☒ | A full description of the statistical parameters including central tendency (e.g. means) or other basic estimates (e.g. regression coefficient) AND variation (e.g. standard deviation) or associated estimates of uncertainty (e.g. confidence intervals) |
| ☒ | ☐ | For null hypothesis testing, the test statistic (e.g. *F*, *t*, *r*) with confidence intervals, effect sizes, degrees of freedom and *P* value noted *Give P values as exact values whenever suitable.* |
| ☒ | ☐ | For Bayesian analysis, information on the choice of priors and Markov chain Monte Carlo settings |
| ☒ | ☐ | For hierarchical and complex designs, identification of the appropriate level for tests and full reporting of outcomes |
| ☒ | ☐ | Estimates of effect sizes (e.g. Cohen's *d*, Pearson's *r*), indicating how they were calculated |

*Our web collection on statistics for biologists contains articles on many of the points above.*

## Software and code

Policy information about availability of computer code

| Data collection | Zeiss ZEN 3.2 blue edition, xT microscope control software v6.2.6. For the acquisition of raw mass spectrometric and raw sequencing data, instrument-supplied software was utilized. |
|---|---|
| Data analysis | Matlab 2018b (Mathworks), QIIME2 (and the following plugins: cutadapt v1.16, VSEARCH 2.14.1, Deblur 1.1.0, q2-feature-classifier 2020.2.0.dev0, alpha-group-significance, DEICODE v0.2.3, q2-emperor 2020.2.0.dev0, beta-group-significance), Songbird v1.0.0, Qurro v0.5.0., phyloseq v1.22.3, phyloFlash 3.0/3.3b3, bbmap v38.75, metaFlye version 2.7, SAMtools version 1.10, Pilon version 1.23, CheckM version 1.0.18, Prokka 1.13.3, BWA-0.7.17, minimap2-2.17, dbCAN meta server (incl. HMMER, DIAMOND and Hotpep), RAST annotation server, SINA aligner 1.2.11, ARB 6.1, Trimmomatic 0.32, SortMeRNA 2.1, Bowtie2 2.1.0, samtools 0.1.19, featureCounts 1.4.6, BRIG v0.95, look@NanoSims v2018, Microsoft Excel. Custom codes for processing of eddy correlation data and nanoSIMS data can be found at github under https://github.com/SoerenAhmerkamp/EddyCorrelation and https://github.com/SoerenAhmerkamp/NanoSIMS/. |

For manuscripts utilizing custom algorithms or software that are central to the research but not yet described in published literature, software must be made available to editors and reviewers. We strongly encourage code deposition in a community repository (e.g. GitHub). See the Nature Portfolio guidelines for submitting code & software for further information.

## Data

Policy information about availability of data

All manuscripts must include a data availability statement. This statement should provide the following information, where applicable:
- Accession codes, unique identifiers, or web links for publicly available datasets
- A description of any restrictions on data availability
- For clinical datasets or third party data, please ensure that the statement adheres to our policy

Raw reads of the 16S rRNA gene amplicon sequencing, the MAGs of C. neptuna and C. diazotrophica (DSM18577), and the mapped reads of the transcriptomes are available under Bioproject number PRJEB37438 at the European Nucleotide Archive (ENA). Sequences that were included in the phylogenetic tree are available in Supplementary File 1 (with accession numbers and references) and as a tree file (Supplementary File 2). The comparison of 34 genomes for presence/absence of specific genes and/or pathways is available in Supplementary File 3 including their accession numbers. The PhyloFlash results (as presented in Extended Data Fig. 2) are available in Supplementary File 4. Publicly available sequences used for phylogenetic tree construction and genome comparison can be found under their respective accession numbers at NCBI (https://www.ncbi.nlm.nih.gov/) or ENA (https://www.ebi.ac.uk/ena/browser/home). Ribosomal subunit databases used for taxonomic classification can be found at the SILVA rRNA database (https://www.arb-silva.de/). Source data are provided with this paper.

# Field-specific reporting

Please select the one below that is the best fit for your research. If you are not sure, read the appropriate sections before making your selection.

☐ Life sciences ☐ Behavioural & social sciences ☒ Ecological, evolutionary & environmental sciences

For a reference copy of the document with all sections, see nature.com/documents/nr-reporting-summary-flat.pdf

# Ecological, evolutionary & environmental sciences study design

All studies must disclose on these points even when the disclosure is negative.

| | |
|---|---|
| Study description | We carried out several field sampling campaigns to Posidonia oceanica meadows in the Mediterranean Sea. Sampling and experiments were set up to i) measure in situ rates of primary production and O2 penetration into the sediment, ii) measure N2 fixation rates associated with the roots of the seagrass and the subsequent transfer of freshly fixed N, iii) to subsample these rate incubations to visualize N2-fixing microorganisms, and iv) to study their potential metabolism via sequencing of metagenomes and metatranscriptomes. This study was designed to obtain a mechanistic insight into the interactions of N2-fixing microorganisms with the P. oceanica plant. |
| Research sample | The seagrass Posidonia oceanica is one of the most prolific seagrasses, producing large amounts of biomass. Its growth in a nutrient-poor environment indicates that microbial N2 fixation is important for this ecosystem. Our prior work showed that N2 fixation was mostly associated with the roots of the seagrass. We therefore focused on studying the root-associated N2 fixation activity as well as the transfer of the newly fixed N to the leaves. Sediment, pore water and the overlying water column were also sampled to obtain environmental data relevant to our study. Fetovaia Bay (Elba, Italy) was chosen as a study site due to its pristine nature and the oligotrophic conditions representative of Posidonia meadows. Further, the study site is near a field station with sampling and laboratory infrastructure, which allowed repeated sampling and the completion of experiments. |
| Sampling strategy | The manuscript reports data from several field campaigns across different seasons and years. Sampling size was largely determined by the feasibility of experiments with triplicate incubated plants for each set of experiments and at least one experiment per sampling campaign. The three replicate plants for each experimental set is warranted by the various sampling campaigns across different seasons. |
| Data collection | Eddy correlation measurements were carried out in situ during 13-24 hour deployments. Diving staff of Hydra Marine Sciences, Hydra Field work and Soeren Ahmerkamp were present during the deployment and recovery of the instrumentation (high-frequency current meters combined with fast O2 microsensors; sediment microsensors). Water column and sediment O2 concentrations were taken automated during this deployment time. Plant and sediment samples were collected by staff of Hydra Marine Sciences and Hydra Field Work. Nutrient data (fluorometrically/photometrically) was collected either in the nearby laboratory on Elba Island, Italy, (collected by Nadine Lehnen, Hannah Marchant, Wiebke Mohr and/or technical support staff) or using an autoanalyzer at the Max Planck Institute Bremen (operated by technical support staff). Biomass and mass spectrometric data was collected using an elemental analyzer coupled to a continuous-flow isotope ratio mass spectrometer (equipped with an autosampler, operated by technical support staff), a nanoscale secondary ion mass spectrometer (operated by Sten Littmann and technical support staff), a gas chromatograph and a gas chromatograph coupled to an isotope ratio mass spectrometer (operated by technical support staff; Eawag). Sequencing data was collected at the Max Planck Genome Centre Cologne (individual sequencing platforms are detailed in the methods section). |
| Timing and spatial scale | Field campaigns took place in June 2014, May 2015, April 2016, August 2016, May 2017, September 2018, June and September 2019. Different seasons were chosen to observe changes in microbial community and processes with a change from conditions where nutrients are available to nutrient-deplete conditions. The same seagrass meadow was visited during these campaigns (Fetovaia Bay, Elba), and an additional meadow at a different island (Cala della Ruta, Pianosa) was visited once in May 2017.<br>Fetovaia Bay opens easterly/southeasterly to the Mediterranean Sea with a maximum North-South and West-East extent of ~700 m. Cala della Ruta Bay opens southerly to the Mediterranean and has a maximum North-South and West-East extent of about 400 and 700 m, respectively. |

| Data exclusions | We excluded the sequencing data of three plants from the microbial community analyses (16S rRNA gene amplicon sequencing) because they did not pass our initial quality assessment (described in methods). These three plants belonged the largest group of plants for one (non-N2-fixing) season (May). |
|---|---|
| Reproducibility | All experiments described in our study are individual environmental sampling campaigns at different times and/or different years. Within each set of experiments, triplicate incubations and multiple measurements within each triplicate (where applicable) were performed to assess variability. Variability within triplicates was substantial (see Fig. 1, 2 and Extended Data Fig. 3) reflecting biological differences between individual plants and plant pieces rather than errors in measurements. All attempts at replication were successful. |
| Randomization | The plants were randomly collected by divers in accessible spots and considering to minimize potential damage to the seagrass meadow. Incubation plants were randomly chosen from the pool of sampled plants making sure that each incubated plant had sufficient root, rhizome and leaf material for subsequent measurements. |
| Blinding | Blinding was not pertinent to our study because it did not include any animals and/or human research participants. In addition, blinding was not possible since many analyses were also carried out by the persons in charge of sampling and interpretation of the data was done by persons in charge of analyses. |

Did the study involve field work? ☒ Yes ☐ No

## Field work, collection and transport

| Field conditions | Weather conditions were calm enough to allow SCUBA diving and sample collection. Water temperatures were 16 °C (April), 18 °C (May 2015 and 2017), 23 °C (June 2014), 24 °C (June 2019), 26 °C (August 2016), 25 °C (September 2018) and 22 °C (September 2019). Nutrient concentrations in the water column above the seagrass meadow and in porewaters of seagrass and neighboring sediments were measured during several campaigns and are detailed in Extended Data Table 1. |
|---|---|
| Location | Posidonia oceanica meadow and neighboring sediments in Fetovaia Bay, Elba Island (Italy; 42°43.804'N 10°09.422'E) and Posidonia oceanica meadow in Cala della Ruta Bay, Pianosa Island (Italy; 42°34.362'N 010°03.795'E). Sampled meadows and sediments are at water depth between 5-10 m. |
| Access & import/export | The Posidonia oceanica meadows were accessed using minimal-invasive zodiacs/small boats and SCUBA-diving/snorkling. Local, national and international laws were followed. Access to the protected waters of the Island of Pianosa was granted by the National Park Tuscan Archipelago, Portoferraio, Italy (permit no. 2930/2017). |
| Disturbance | Plants were carefully separated from the meadow by hand to minimize damage/disturbance to the ecosystem. In situ measurements were minimally invasive. |

# Reporting for specific materials, systems and methods

We require information from authors about some types of materials, experimental systems and methods used in many studies. Here, indicate whether each material, system or method listed is relevant to your study. If you are not sure if a list item applies to your research, read the appropriate section before selecting a response.

## Materials & experimental systems

| n/a | Involved in the study |
|---|---|
| ☒ | ☐ Antibodies |
| ☒ | ☐ Eukaryotic cell lines |
| ☒ | ☐ Palaeontology and archaeology |
| ☒ | ☐ Animals and other organisms |
| ☒ | ☐ Human research participants |
| ☒ | ☐ Clinical data |
| ☒ | ☐ Dual use research of concern |

## Methods

| n/a | Involved in the study |
|---|---|
| ☒ | ☐ ChIP-seq |
| ☒ | ☐ Flow cytometry |
| ☒ | ☐ MRI-based neuroimaging |

