## [Peer Review File · Nature]

Manuscript Title: Terrestrial-type N-fixing symbiosis between seagrass and a marine bacterium

Reviewer Comments & Author Rebuttals

Reviewer Reports on the Initial Version:

Referee #1 (Remarks to the Author):

This manuscript describes an uncultivated gammaproteobacterium, *Candidatus Celerinatantimonas neptuna*, that lives within seagrass roots and provides the plant with fixed nitrogen. This symbiosis was likely critical to enabling seagrass to colonize nitrogen-poor marine sediments, and therefore may play a key role in a major global carbon sink. Through a mix of ¹⁵N isotope labeling experiments, root imaging, and ¹⁶S amplicon, metagenome and metatranscriptome sequencing the investigators demonstrate that N₂ becomes fixed by *C. neptuna* and transferred to the plant, primarily during the summer growth season. They also characterize the genome of *C. neptuna*, derived from assembly and binning of PacBio and Illumina metagenome data from seagrass root samples, an impressive accomplishment given the high abundance of plant DNA in the samples. The work is extensive and well described and has significant implications for the global carbon cycle as well as for understanding of seagrass evolution. The data are clear and nicely presented, and should appeal to a broad audience of microbiologists, plant biologists and ecologists.

Specific comments:

Lines 22-23: "Particularly amongst land plants, N₂-fixing symbionts are ubiquitous and involve an intimate association between host and symbiont^{2,4}." Is this really accurate that nitrogen-fixing symbioses are "ubiquitous" in plants? Of the two references cited, one is only focused on the rhizobia-legume symbioses and the other describes a nitrogen-fixing bacterium that colonizes grasses. What about other plants? Similarly, line 32 says that "N₂-fixing symbionts allowed early land plants to colonize N-poor soil⁶" but the cited reference only refers to the emergence of legumes, which isn't consistent with nitrogen-fixing symbionts being widespread in other plants. Lines 155-157: I was surprised to see flagellar and pili genes related to motility and attachment cited as hallmarks of an endophytic lifestyle; wouldn't motility be more useful outside the plant? Are these genes actually overrepresented in endophytes as compared to rhizosphere or soil organisms? And does this (along with the chemotaxis and plant cell wall degradation genes) imply that *C. neptuna* is acquired from the environment rather than vertically inherited?

Lines 195-198: "Interestingly, the potential to fix N₂ is prevalent among the members of the family Celerinatantimonadaceae, while it is missing in the neighboring families Idiomarinaceae and Colwelliaceae (≥85% identity of 16S rRNA gene) (Suppl. File 3), which typically do not associate with macrophytes." How many isolates have been characterized and/or sequenced from these families? It might be more conservative to say it is missing from known representatives of these families.

Minor comments:

Lines 286 – The list of features shown in the image is awkwardly phrased; maybe a word or two is missing?

Referee #2 (Remarks to the Author):

Mohr et al. report on the identification and characterization of an endophytic gammaproteobacterium which inhabits the roots of the seagrass *Posidonia*. They provide a very nice data using state of the art procedures including specific FISH probes to quantify the high density occurrence of this bacterium in *Posidonia* roots and transcriptomics to examine enzyme

expression profiles consistent with a diazotroph. Tracer $^{15}\text{N}_2$ isotope uptake is employed to estimate rates of uptake by the whole plant, to localize its movement from symbiont to plant using nanoSIMS and also to demonstrate translocation from the roots to the leaves as well as the enrichment in the amino acid fraction. Using metagenomics and transcriptomics, they provide some speculation on the possible underlying interactions of the symbiosis, including metabolite exchange and other mechanisms to potentially foster nitrogen fixation- paralleling the well known terrestrial legume-rhizobial systems. They also speculate on the evolution and development of the symbiosis for seagrasses which had themselves terrestrial origins and re-invaded marine habitats. One thing unclear to this reviewer is whether the bacterium was actually isolated and grown- which does not seem to be the case. Also, some of the inferences regarding the symbiosis are relatively speculative- e.g. the role of the plant in providing carbon skeletons and the microaerophilic nature of the relationship. Has the candidate bacterium been established as a microaerophile?

Nonetheless, this is a very interesting study which advances the understanding of how seagrass communities thrive in otherwise nutrient depleted warm water environments.

Specific comments:

Line 24 -25. In contrast to this statement, two prior isolations of bacteria from the roots of *Zostera* are reported in Table 1 of Flynn 2000 although the extent of characterization was not detailed there.

Line 77-80. Capone earlier reported on the transfer of $^{15}\text{N}_2$ from roots to leaves in *Zostera marina*. (Capone, D. G. 1988. Benthic nitrogen fixation. In T. H. Blackburn & J. Sorensen (Eds.), Nitrogen cycling in coastal marine environments. (pp. 85-123). New York: J. Wiley & Sons.)

Lines 83-85. These are not very high concentrations of NH_4 for a sediment and could also derive from decomposition and DNRA reactions. I assume these are relatively coarse grained, oxic sediments.

Lines 141-153. This seems reasonable but at this point is pretty speculative. Again, this may go to the nature of the sediments – whether they are fine grained and organic rich and prone to anoxia or coarse grained. Seagrasses can thrive in both and the nature of the microbial communities will be affected.

Douglas G. Capone

Referee #3 (Remarks to the Author):

This study documents the presence of N_2 fixing bacteria in the roots in the seagrass *P. oenocarpa*. The study is novel by documenting that the bacteria are found in the roots, whereas N_2 fixing activity until now primarily has been documented in the rhizosphere sediments. The study is impressive in the documentation of the bacteria and their role in the seagrass ecosystem. The study covers almost the entire ecosystem from the overall productivity in the seagrasses to the smallest scale of genes. The approach is remarkable by covering so many aspects of the N_2 fixing in seagrasses and the findings are carefully supported by measuring and documenting the mechanisms, processes and organisms etc. The replication of individual measurements, e.g. the number of seagrass cores from where leaves for incubations were selected, is not high, but this is fully compensated by addressing aspects like seasonality, which is essential to understand the role of N_2 fixation. Future studies can go more into details on replication, spatial scales and other aspects. The statistics are well designed and described. I have a few minor comments to the figures – see below.

The conclusions are well documented by the thorough analysis as mentioned above. The analysis supports the arguments for the role of N₂ fixation in *P. oceanica* roots during summer months when N resources in the water column are depleted. The possible mechanisms for obtaining the sugars for bacterial growth are explained and mass balance calculations have been done. The extensive molecular work supports the hypothesis on that the mechanism in seagrass roots are similar to terrestrial plants. The microscope and imaging work shows the presence of the bacteria. The list of references is comprehensive and includes credits to the relevant work. The manuscript is very well written and has an excellent flow throughout. The arguments are well aligned and provides a continuous and comprehensive documentation of the findings in this study.

Minor comment

Fig. 1

l. 225 – what do you mean by separate measurements in Fig 1b?

Author Rebuttals to Initial Comments:

Referee #1 (Remarks to the Author):

*This manuscript describes an uncultivated gammaproteobacterium, Candidatus Celerinatantimonas neptuna, that lives within seagrass roots and provides the plant with fixed nitrogen. This symbiosis was likely critical to enabling seagrass to colonize nitrogen-poor marine sediments, and therefore may play a key role in a major global carbon sink. Through a mix of ¹⁵N isotope labeling experiments, root imaging, and ¹⁶S amplicon, metagenome and metatranscriptome sequencing the investigators demonstrate that N₂ becomes fixed by *C. neptuna* and transferred to the plant, primarily during the summer growth season. They also characterize the genome of *C. neptuna*, derived from assembly and binning of PacBio and Illumina metagenome data from seagrass root samples, an impressive accomplishment given the high abundance of plant DNA in the samples. The work is extensive and well described and has significant implications for the global carbon cycle as well as for understanding of seagrass evolution. The data are clear and nicely presented, and should appeal to a broad audience of microbiologists, plant biologists and ecologists.*

We thank the reviewer for the positive feedback and constructive comments.

Specific comments:

Lines 22-23: “Particularly amongst land plants, N₂-fixing symbionts are ubiquitous and involve an intimate association between host and symbiont^{2,4}.” Is this really accurate that nitrogen-fixing symbioses are “ubiquitous” in plants? Of the two references cited, one is only focused on the rhizobia-legume symbioses and the other describes a nitrogen-fixing bacterium that colonizes grasses. What about other plants? Similarly, line 32 says that “N₂-fixing symbionts allowed early land plants to colonize N-poor soil⁶” but the cited reference only refers to the emergence of legumes, which isn’t consistent with nitrogen-fixing symbionts being widespread in other plants.

We agree that the word ‘ubiquitous’ is not accurate here because we were actually referring to the fact that N₂-fixing symbionts occur in a variety of distantly related plants. We have revised our sentence as follows, “Particularly amongst land plants, N₂-fixing symbionts occur in a variety of distantly related plant lineages”

Reviewer 1 is correct; the cited reference (Wang et al., 2020) refers only to the emergence of legumes and as such is not appropriate for this statement. We have replaced this reference

here and at the end of the main text by Knack et al., (2015), who argue that N₂-fixing microorganisms were key to the colonization of land by early plants.

Lines 155-157: I was surprised to see flagellar and pili genes related to motility and attachment cited as hallmarks of an endophytic lifestyle; wouldn't motility be more useful outside the plant? Are these genes actually overrepresented in endophytes as compared to rhizosphere or soil organisms? And does this (along with the chemotaxis and plant cell wall degradation genes) imply that C. neptuna is acquired from the environment rather than vertically inherited?

Based on the literature it is unclear whether flagellar and/or pili genes are overrepresented in endophytes, and published genomic comparisons are inconclusive (Hardoim et al. 2008). Nonetheless, we agree with Reviewer 1 that motility might be more useful outside the plant and hence flagellar and pili genes are not hallmarks of obligate endophytes. Therefore, we now propose that the possession of motility genes, together with the chemotaxis and plant cell wall degradation genes rather indicates that *C. neptuna* is a facultative endophyte that switches between free-living and host-associated stages. Such lifestyle is typical for many terrestrial plant N₂-fixing symbionts. The corresponding section in the manuscript was now rewritten as follows: "In many aspects, the genome of *C. neptuna* exhibits hallmarks of a facultative endophytic symbiont. Just like many terrestrial plant endophytes, *C. neptuna* might switch between free-living and host-associated stages (Hardoim et al. 2008)."

Lines 195-198: "Interestingly, the potential to fix N₂ is prevalent among the members of the family Celerinatantimonadaceae, while it is missing in the neighboring families Idiomarinaceae and Colwelliaceae (≥85% identity of 16S rRNA gene) (Suppl. File 3), which typically do not associate with macrophytes." How many isolates have been characterized and/or sequenced from these families? It might be more conservative to say it is missing from known representatives of these families.

Until now, twenty-eight genomes in both the Colwelliaceae (14) and the Idiomarinaceae (14) have been sequenced (summarized in Supplementary File 3). We have revised the sentence according to the suggestion of reviewer 1.

Minor comments:

Lines 286 – The list of features shown in the image is awkwardly phrased; maybe a word or two is missing?

We have rephrased the sentence in the revised manuscript.

Referee #2 (Remarks to the Author):

Mohr et al. report on the identification and characterization of an endophytic gammaproteobacterium which inhabits the roots of the seagrass Posidonia. They provide a very nice data using state of the art procedures including specific FISH probes to quantify the high density occurrence of this bacterium in Posidonia roots and transcriptomics to examine enzyme expression profiles consistent with a diazotroph. Tracer 15N₂ isotope uptake is employed to estimate rates of uptake by the whole plant, to localize it's movement from symbiont to plant using nanoSIMS and also to demonstrate translocation from the roots to

the leaves as well as the enrichment in the amino acid fraction. Using metagenomics and transcriptomics, they provide some speculation on the possible underlying interactions of the symbiosis, including metabolite exchange and other mechanisms to potentially foster nitrogen fixation- paralleling the well known terrestrial legume-rhizobial systems. They also speculate on the evolution and development of the symbiosis for seagrasses which had themselves terrestrial origins and re-invaded marine habitats.

We thank the reviewer for the positive feedback and constructive comments.

One thing unclear to this reviewer is whether the bacterium was actually isolated and grown- which does not seem to be the case. Also, some of the inferences regarding the symbiosis are relatively speculative- e.g. the role of the plant in providing carbon skeletons and the microaerophilic nature of the relationship. Has the candidate bacterium been established as a microaerophile?

To clarify, we have not yet isolated the bacterium; instead, the physiology of this organism is inferred from genome and transcriptome data. Therefore, we have now toned down the corresponding statements in the discussion.

We did not mean to imply that the bacterium is a microaerophile, albeit, based on the presence of both high- and low-affinity terminal oxidases in its genome, we believe it can live under fully oxic as well as microoxic conditions. We proposed that the bacterium is growing under microoxic and/or anoxic conditions in the roots based on the low transcription of the low O₂-affinity terminal oxidase, high transcription of the high O₂-affinity terminal oxidase and fermentation genes, as well as the anoxic nature of the sediment surrounding the roots. We have now included this in the revised manuscript. Additionally, we added microsensor data showing that the seagrass meadow sediments are largely anoxic with O₂ penetrating to a depth of about 4 mm, similar to observations by Holmer et al. (2003).

Nonetheless, this is a very interesting study which advances the understanding of how seagrass communities thrive in otherwise nutrient depleted warm water environments.

Specific comments:

Line 24 -25. In contrast to this statement, two prior isolations of bacteria from the roots of Zostera are reported in Table 1 of Flynn 2000 although the extent of characterization was not detailed there.

Indeed, a few bacteria have been isolated from the roots of seagrasses but the nature of their association (e.g. pathogenic vs. beneficial, endophyte vs. epiphyte) with seagrasses is so far unknown. It is generally assumed that seagrasses only form loose associations with bacteria, and the kind of specific intimate beneficial N₂-fixing symbiosis that we report here has not been described for seagrasses before. To clarify, we have revised our sentence as follows: “Such intimate symbioses have so far not been described for seagrasses....”

Line 77-80. Capone earlier reported on the transfer of 15N₂ from roots to leaves in Zostera marina. (Capone, D. G. 1988. Benthic nitrogen fixation. In T. H. Blackburn & J. Sorensen (Eds.), Nitrogen cycling in coastal marine environments. (pp. 85-123). New York: J. Wiley & Sons.)

We have included the following sentence in the revised manuscript: “Such rapid transfer to the leaves was previously reported for *Zostera marina* (Capone 1988).”

Lines 83-85. These are not very high concentrations of NH₄ for a sediment and could also derive from decomposition and DNRA reactions. I assume these are relatively coarse grained, oxic sediments.

We agree that these ammonium concentrations are not very high for sediments. It is unlikely that DNRA is a substantial source of ammonium due to the low to non-detectable NO_x⁻ concentrations in the porewater/seawater. Due to non-detectable concentrations in the seawater, inorganic N cannot be the N-source for organic matter synthesis in summer. Hence N₂ fixation is essentially the only relevant source of N for the wider seagrass ecosystem in summer and thus indirectly also of ammonium released upon organic matter decomposition. For question about sediment properties, please see answer below.

Lines 141-153. This seems reasonable but at this point is pretty speculative. Again, this may go to the nature of the sediments – whether they are fine grained and organic rich and prone to anoxia or coarse grained. Seagrasses can thrive in both and the nature of the microbial communities will be affected.

As mentioned in our answer to your earlier comment, the physiological traits described in this particular section are inferred from genome and transcriptome data, as well as biogeochemical in situ measurements. This section has been toned down and partly rephrased (see above).

Concerning the sediment properties: At our study site, the sediments are characterized as fine sands with average grain sizes of ~170 μm. *Posidonia* meadow sediments are relatively organic-rich (Holmer et al., AME, 2004) and only the upper few millimeters of these sediments are oxic with the *P. oceanica* roots residing in anoxic sediments (e.g. Holmer et al. 2003). We have now included own microsensor measurements that show that the sediment at our site are also largely anoxic. We agree that these physico-chemical properties will affect the composition of the microbial community.

Douglas G. Capone

Referee #3 (Remarks to the Author):

*This study documents the presence of N₂ fixing bacteria in the roots in the seagrass *P. oceanica*.*

The study is novel by documenting that the bacteria are found in the roots, whereas N₂ fixing activity until now primarily has been documented in the rhizosphere sediments

The study is impressive in the documentation of the bacteria and their role in the seagrass ecosystem. The study covers almost the entire ecosystem from the overall productivity in the seagrasses to the smallest scale of genes. The approach is remarkable by covering so many aspects of the N₂ fixing in seagrasses and the findings are carefully supported by measuring and documenting the mechanisms, processes and organisms etc. The replication of individual measurements, e.g. the number of seagrass cores from where leaves for incubations were selected, is not high, but this is fully compensated by addressing aspects like seasonality,

which is essential to understand the role of N₂ fixation. Future studies can go more into details on replication, spatial scales and other aspects.

The statistics are well designed and described. I have a few minor comments to the figures – see below.

*The conclusions are well documented by the thorough analysis as mentioned above. The analysis supports the arguments for the role of N₂ fixation in *P. oceanica* roots during summer months when N resources in the water column are depleted. The possible mechanisms for obtaining the sugars for bacterial growth are explained and mass balance calculations have been done. The extensive molecular work supports the hypothesis on that the mechanism in seagrass roots are similar to terrestrial plants. The microscope and imaging work shows the presence of the bacteria.*

The list of references is comprehensive and includes credits to the relevant work

The manuscript is very well written and has an excellent flow throughout. The arguments are well aligned and provides a continuous and comprehensive documentation of the findings in this study.

We thank the reviewer for the positive feedback and constructive comments.

Minor comment

Fig. 1 l. 225 – what do you mean by separate measurements in Fig 1b?

Each of the four bars presented in Fig. 1b is an individual 24-hour measurement. We have revised the figure legend to read, “The four individual bars represent means and standard deviations of averaged hourly fluxes recorded over the course of 24h. The four measurements (two seagrass and two sand) were performed on four different days (see methods and Extended Data Fig. 1).”

Reviewer Reports on the First Revision:

Referee #1 (Remarks to the Author):

I have looked over the responses and the revisions and all of my concerns have been addressed.

Referee #2 (Remarks to the Author):

This is a re-review and I am satisfied with the authors response to my comments.